# Multiple functions of cerebello-thalamic neurons in learning and offline consolidation of a motor skill in mice

**Andrés Pablo Varani‡, Caroline Mailhes-Hamon, Romain W Sala, Marie Sarraudy, Sarah Fouda, Jimena L Frontera, Clément Léna*‡, Daniela Popa*‡**

Institut de biologie de l'Ecole normale supérieure (IBENS), Ecole normale supérieure, CNRS, INSERM, PSL Research University, Paris, France

**\*For correspondence:**
lena@biologie.ens.fr (CL);
dpopa@biologie.ens.fr (DP)

**Present address:** ‡Universidad de Buenos Aires - CONICET. Instituto de Fisiología y Biofísica Bernardo Houssay (IFIBIO Houssay), Facultad de Medicina, Departamento de Ciencias Fisiológicas. Grupo de Neurociencias de Sistemas, Buenos Aires, Argentina

‡These authors jointly directed the work

**Competing interest:** The authors declare that no competing interests exist.

## eLife Assessment

Varani et al present **important** findings regarding the role of distinct cerebellothalamic connections in motor learning and performance. The evidence supporting the main claims is **convincing**, with multiple replications, validation of their techniques, and appropriate controls. The work will be of broad interest to neuroscientists interested in central mechanisms of motor learning and control, as well as thalamic physiology.

**Abstract** Motor skill learning is a complex and gradual process that involves the cortex and basal ganglia, both crucial for the acquisition and long-term retention of skills. The cerebellum, which rapidly learns to adjust the movement, connects to the motor cortex and the striatum primarily via the ventral and intralaminar thalamus, respectively. Here, we evaluated the contribution of cerebellar neurons projecting to these thalamic nuclei in a skilled locomotion task in mice. Using a targeted chemogenetic inhibition that preserves the motor abilities, we found that cerebellar nuclei neurons projecting to the intralaminar thalamus contribute to learning and expression, while cerebellar nuclei neurons projecting to the ventral thalamus contribute to offline consolidation. Asymptotic performance, however, required each type of neurons. Thus, our results show that cerebellar neurons belonging to two parallel cerebello-thalamic pathways play distinct, but complementary, roles functioning on different timescales and both necessary for motor skill learning.

## Introduction

Learning to execute and automatize certain actions is essential for survival and, indeed, animals have the ability to learn complex patterns of movement with great accuracy to improve the outcomes of their actions (*Krakauer et al., 2019*). Two categories of learning are often engaged in motor skill acquisition (*Seidler, 2010*; *Krakauer et al., 2019*): (1) sequence learning, which is needed when series of distinct actions are required, and (2) adaptation, which corresponds to learning a variation of a previous competence and typically takes place when motor actions yield unexpected sensory outcomes. The neurobiological substrate of motor skills involves neurons distributed in the cortex, basal ganglia, and cerebellum, each structure using different learning algorithms (*Doya, 1999*). On one hand, the cerebellum is a site where supervised learning takes place (*Raymond and Medina, 2018*). The cerebellum is thought to form associations between actions and predicted sensory outcome at short-time scale (typically under one second), which are seen as internal models (*Ito, 2008*) and are essential for the precise timing and coordination required for motor skill execution. The cerebellum has been shown to be central for the adaptation of skills such as oculomotor movements (*Nguyen-Vu et al., 2013*; *Yang*

*and Lisberger, 2014*; *Herzfeld et al., 2018*), reaching (*Hewitt et al., 2015*), locomotion (*Morton and Bastian, 2006*; *Darmohray et al., 2019*), as well as conditioned reflexes (*Clopath et al., 2014*; *Longley and Yeo, 2014*). On the other hand, reinforced learning takes place in the basal ganglia and is required to learn complex actions involving sequences of movements, for which the involvement of the cerebellum is much less understood (*Seidler et al., 2002*; *Bernard and Seidler, 2013*; *Krakauer et al., 2019*; *Baetens et al., 2020*).

In general, motor skills are progressively acquired (e.g. *Karni et al., 1998*) and their execution may ultimately recruit sets of brain structures distinct from those involved in the initial training (e.g. *Brashers-Krug et al., 1996*; *Muellbacher et al., 2002*; *Korman et al., 2003*). Strikingly, motor skill consolidation does not occur simply 'online', during repeated task execution, but also 'offline' during the resting periods (*Brashers-Krug et al., 1996*; *Muellbacher et al., 2002*; *Cohen et al., 2005*; *Doyon et al., 2009*). Indeed, a single resting period may be sufficient to change the brain regions recruited in task execution (*Shadmehr and Holcomb, 1997*). Moreover, motor memories may also persist in the form of 'savings', which facilitate relearning of the task at a later time (*Huang et al., 2011*; *Mauk et al., 2014*). Overall, motor skill learning is a dynamical process distributed in time and across brain regions.

Comprehending the role of the cerebellum in motor skill learning, beyond its role in motor coordination, requires considering its integration in brain-scale circuits including the cortex and basal ganglia (*Caligiore et al., 2017*). In the mammalian brain, both cerebellum and basal ganglia receive the majority of their inputs from the cerebral cortex, via the pontine nuclei for the cerebellum. The cerebellum and basal ganglia send projections back to the cortex via anatomically and functionally segregated channels, which are relayed by predominantly non-overlapping thalamic regions (*Bostan et al., 2013*; *Proville et al., 2014*; *Hintzen et al., 2018*). Furthermore, anatomical and functional reciprocal di-synaptic connections have been described between the basal ganglia and the cerebellum (*Bostan and Strick, 2010*; *Carta et al., 2019*). Cerebellar projections to the striatum and to the motor cortex are relayed primarily through distinct thalamic regions, respectively the intralaminar thalamus and ventral thalamus (*Steriade, 1995*; *Chen et al., 2014*), suggesting distinct contributions of these cerebello-diencephalic projections.

In the present study, we hypothesized that the cerebellum may contribute to some phases of learning in a complex motor task via its projections to thalamic nuclei embedded in thalamo-cortical and thalamo-striatal networks. We studied the accelerating rotarod task, which learning depends on the cortex and basal ganglia (*Costa et al., 2004*; *Cao et al., 2015*; *Kida et al., 2016*). We then focused on the cerebellar nuclei (CN) and their projections to the centrolateral (intralaminar) thalamus and ventral anterior lateral complex (motor thalamus), which are known to relay their activity to the striatum and the motor cortex (*Chen et al., 2014*; *Proville et al., 2014*; *Gornati et al., 2018*), and whose inhibition does not impair the ability to walk on a rotating rod. We examined the contribution of CN and CN neurons belonging to distinct cerebello-thalamic pathways to learning using chemogenetic disruptions either during or after the learning sessions and revealed a differential contribution of the distinct CN-thalamic neurons to the online learning and the offline consolidation.

## Results

In the present study, we used the paradigm of the accelerating rotarod, in which the animals walk on an accelerating rotating horizontal rod. Over multiple repetitions of the tasks, the rodents develop locomotion skills to avoid falling from the rod. This paradigm allows studying the neurobiological basis of motor skill learning (e.g. *Costa et al., 2004*; *Yang et al., 2009*; *Rothwell et al., 2014*), especially at multiple time scales. When repeated over multiple days, distinct phases of learning, with different rate of performance improvement and organization of locomotion strategies, can be distinguished (*Buitrago et al., 2004*) and selectively disrupted (*Hirata et al., 2016*; *Sathyamurthy et al., 2020*).

### Partial inhibition of the CN neurons using hM4D(Gi)-DREADD does not impair basic motor abilities

To evaluate the contribution of the cerebellar outputs during and after the accelerating rotarod, we employed a chemogenetic approach (*Roth, 2016*) using the inhibitory DREADD (hM4D-Gi) activated by the synthetic drug Clozapine-N-Oxide (CNO).

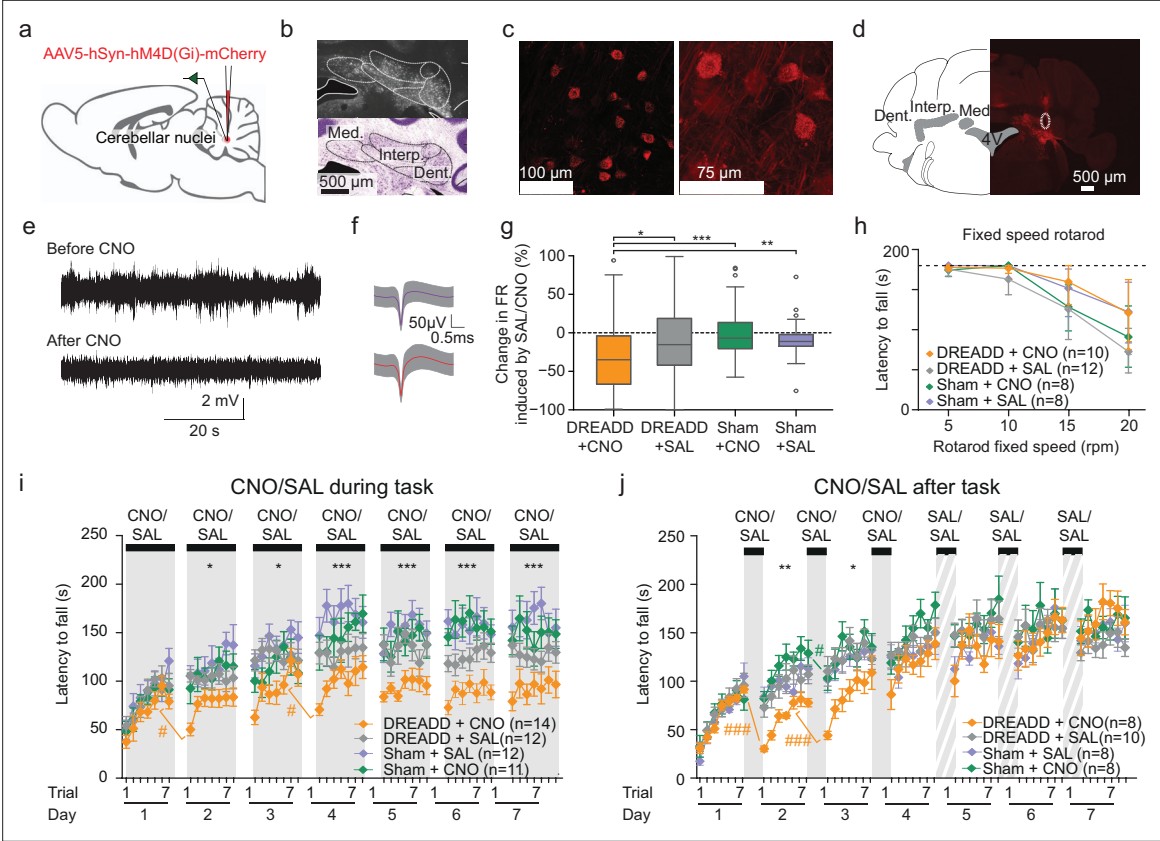

**Figure 1.** Cerebellar nuclei partial inhibition during and after a motor task impairs learning. (**a**) Scheme of the implantation and injection. (**b**) Coronal section of the cerebellum showing hM4Di-DREADD expression in the three CN. (**c**) Representative confocal image showing hM4Di-DREADD expression on neuronal membranes. (**d**) Electrode placement close to cells expressing hM4Di-DREADD (red: lesion site, yellow: electrode track). (**e**) Examples of high-pass filtered traces of CN recordings before and after CNO injection. (**f**) Examples of spike shapes obtained from spike sorting in CN (line + shading indicates mean +/- SD). (**g**) Boxplots displaying the percentage of modulation of CN neurons average firing rate induced by CNO or SAL injections during an open field session. CN firing rate was reduced after 1 mg/kg of CNO injection in DREADD-injected mice compared to other groups (Wilcoxon test* p<0.05, ***p<0.001. Boxes represent quartiles and whiskers correspond to range; points are singled as outliers if they deviate more than 1.5 x interquartile range from the nearest quartile). (**h**) Latency to fall during fixed speed rotarod (5, 10, 15, 20 r.p.m.) for all experimental groups. One way repeated-measure ANOVA on averaged values for all the speed steps in each experimental group followed by Tukey Post hoc pairwise comparison (p>0.05 in all cases). (**i**) Impact of daily injections of CNO before trial 1 on accelerating rotarod performance. Summary of the performance for each trial/ day (repeated measure ANOVA Group effect *p<0.05, ***p<0.001; # p<0.05, ###*P*p<0.001 Tukey pairwise test last trial of each day vs first trial of next day).(**j**) Impact of daily injections of CNO after the task (30 min after trial 7). All treatment is switched to saline in the Late phase. Same presentation as in (**i**). Data in h,i,j are presented as mean ± S.E.M. *n* indicates the number of mice.

The online version of this article includes the following figure supplement(s) for figure 1:

**Figure supplement 1.** DAPI-positive neurons expressing hM4Di-DREADD-mCherry in cerebellar nuclei.

**Figure supplement 2.** Cerebellar nuclei inhibition did not affect execution and fatigue, locomotion, motor coordination, balance, and strength.

**Figure supplement 3.** Extracellular recordings in the CN.

In order to validate this approach, mice were injected with AAV5-hSyn-hM4D(Gi)-mCherry (DREADD mice) or AAV5-hSyn-EGFP (Sham mice) and implanted with microelectrode arrays in the CN (*Figure 1A, B and C*). Post-hoc histology confirmed the position of the electrodes (*Figure 1D*) and showed that the expression of hM4D(Gi)-mCherry was confined to the CN and expressed in neurons, with a large proportion of the cells expressing hM4D(Gi)-DREADD in the CN (*Figure 1— figure supplement 1*). A week after surgery, the neuronal activity was recorded in the CN in an open-field arena before and after accelerating rotarod sessions. CNO injection in mice which received DREADD-expressing virus yielded an ~35% decrease in firing rate, which was not observed following saline (SAL) injection or following saline or CNO injection in Sham mice (*Figure 1E, F and G*; Sham mice which received either SAL or CNO are, respectively, referred to as Sham + SAL and Sham + CNO

while DREADD mice which received SAL or CNO are referred to as DREADD + SAL and DREADD + CNO; detailed statistics in *Supplementary file 1*).

We then examined whether the reduction of CN firing impacted the motor function. We first assessed spontaneous motor activity, strength, and motor coordination. No significant differences were observed between the experimental groups in footprint patterns (*Figure 1—figure supplement 2*), grid test, horizontal bar, and vertical pole (*Figure 1—figure supplement 2B*, detailed statistics in *Supplementary file 6*) indicating that motor coordination and strength were not affected by the reduction of CN firing induced by the administration of CNO 1 mg/kg. The average velocity of open-field locomotion was not altered by CNO or SAL injection in DREADD or Sham mice (*Figure 1—figure supplement 2C*, detailed statistics in *Supplementary file 7*) consistent with intact basic locomotion skills and motivation. As locomotion on a rotating rod may require specific sensorimotor abilities, a crucial control measure is to confirm that the animals can move normally on a fixed-speed rotarod when not subjected to acceleration challenges. We observed that mice with CN inhibition (DREADD + CNO group) exhibited similar performance to the ones of the control groups (DREADD + Saline, Sham + CNO, Sham + Saline, *Figure 1H*, detailed statistics in *Supplementary file 2*). Thus, these results indicate that the reduction of the CN firing induced by the CNO had a limited impact on the motor abilities of the mice, and it is thus appropriate to examine the cerebellar contribution to motor learning rather than to motor function.

## Partial CN inhibition during or after the task has a different impact on motor learning

To test the effect of a CN activity reduction on motor learning, we first examined the impact of CN inhibition by injecting CNO (1 mg/kg) each day before the first trial of an accelerating rotarod session (*Figure 1I*, detailed statistics in *Supplementary file 3* and *Supplementary file 4*) over 7 days. Most of the performance improvement took place in the first 4 days, referred to as 'Early Phase', whereas stable performances were observed in the last 3 days, here referred to as 'Late phase'.

Inhibition of CN activity during the task (*Figure 1I*, detailed statistics in *Supplementary file 4Supplementary file 3* and *Supplementary file 4*) did not affect significantly the learning of DREADD-expressing mice on the first day of the Early Phase, but reduced their performance on the following days compared to the control groups. During the Early phase, overnight loss of performance was observed in the DREADD + CNO mice between the last trial of one day and the first trial of the following day, indicating an impairment in motor learning consolidation. As the effects of systemic CNO administration last longer than the duration of the trials, the results above do not allow us to distinguish the effect of cerebellar inhibition during the trials, or after training (offline consolidation). We therefore injected another set of mice with CNO after the training sessions (*Figure 1J*, detailed statistics in *Supplementary files 3–5*). CN inhibition after the task in the Early Phase indeed reduced the performance on the first trials of the next day, consistent with a disruption of offline consolidation. The performance of DREADD + CNO mice on the last day of the Early Phase was not different from the control groups, indicating that the lack of offline consolidation was overcome by the training the next day. To test whether the skill consolidated under CNO remained stable thereafter, we then shifted the treatment, removing CNO during the Late phase, and we found that mice did not exhibit any further difference between groups. Overall, these results indicate that the offline CN activity participates in the consolidation of the accelerating rotarod learning.

Accelerating rotarod learning is a cumulative process over multiple trials and multiple days. To disentangle differences in learning from differences in consolidation, we examined the change of performance of single animals within and across days. In order to minimize the effect of inter-trial variability on the estimation of performance and learning, we simplified the data by performing a linear regression on the performance of each day where the slope indicates the daily learning rate (*Figure 2A*). The endpoints at trial 1 and 7 of this regression are used as estimates of the initial and final skill level on each day (*Figure 2B*), allowing us thus to follow the daily learning, overnight changes, and offline consolidated learning (*Figure 2B*). In the control groups, we found an inverse correlation between the initial performance of each day and the amount of daily learning: animals with strong initial performance on a given day showed weaker improvements than animals with poor initial performances (*Figure 2C*, top, detailed statistics in *Supplementary file 8*). This result indicates that the comparison of the daily learning between groups of animals requires taking into account the daily

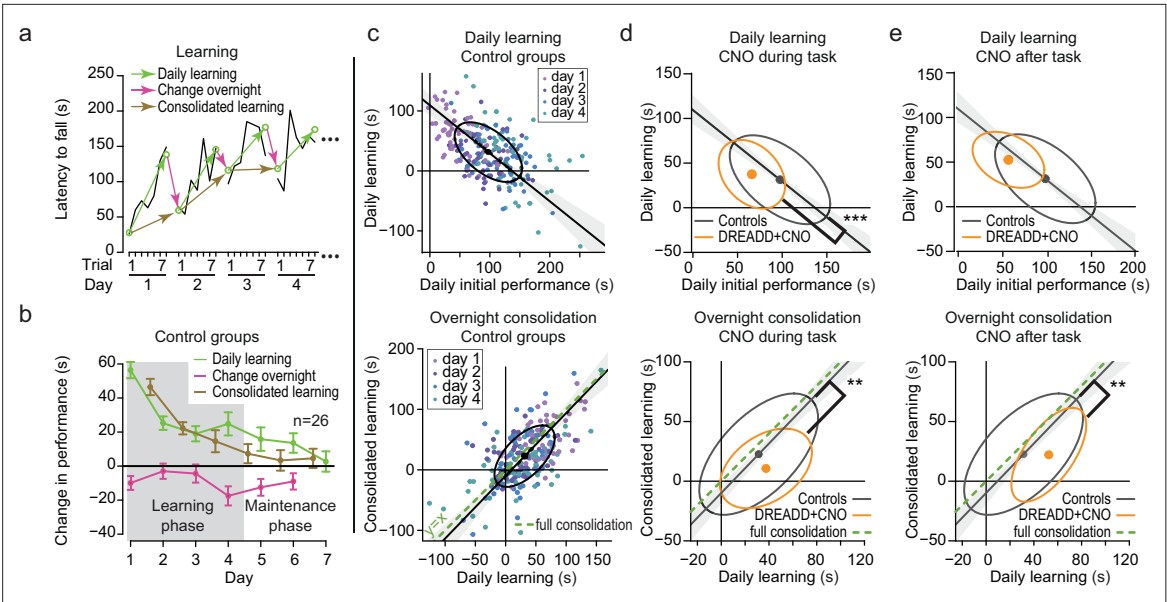

**Figure 2.** Changes in learning and consolidation of motor memory induced by cerebellar nuclei inhibition. (**a**) Example of evolution of latencies to fall for a mouse during the accelerating rotarod protocol. Linear regressions estimated values of trials 1 and 7 during each day are shown using green hollow dots, within-day learning (green arrows), overnight change (red arrows), and day + night learning (brown arrows). (**b**) Evolution of estimated daily learning, overnight change, and consolidated learning shows learning mostly in days 1–4 (Early Phase). Plots represent mean ± S.E.M. (**c**) Within-day learning vs daily initial performances (top) and within-day learning vs consolidated learning (bottom) in the Early phase. Scatterplot of performance from all control mice; the ellipse contains 50% of a bivariate normal distribution fitted to the values and the dot indicates the center of the distribution. Deming linear regression outcomes are represented with 95% confidence interval in shaded color. (**d**) Same as panel (**c**) with superimposition of controls (black) and mice expressing DREADD in DCN and CNO during the task (orange); only ellipses and regression line are included for clarity of the graph (**p<0.01 ***p<0.001 Wilcoxon test for difference between groups of residuals, i.e. signed distance of performances to Deming regression line of control mice). (**e**) Same as panel (**d**) for CNO administered after the task.

initial performance of each animal. In the control mice, daily learning was almost completely preserved overnight, consistent with an effective consolidation across days (*Figure 2C*, bottom, detailed statistics in *Supplementary file 8*). Strikingly, DREADD-expressing mice which received CNO during the task exhibited both a lower learning performance (lower daily increase in performance for similar daily initial performance; *Figure 2D*, top, detailed statistics in *Supplementary file 9*) and lower consolidation (lower consolidated learning for similar daily learning; *Figure 2D* bottom, detailed statistics in *Supplementary file 9*) than control mice. Therefore, the lower performance of these DREADD-expressing mice receiving CNO during the task was not due solely to a disrupted consolidation. In contrast, DREADD-expressing mice, which received CNO after the task, showed normal learning performance (i.e. similar daily increase in performance for similar daily initial performance) (*Figure 2E*, top, detailed statistics in *Supplementary file 9*) but lower consolidation (*Figure 2E*, bottom, detailed statistics in *Supplementary file 9*).

## Selective inhibition of cerebellar neurons projecting to distinct thalamic regions differentially impacts motor learning

Since the CN project to a wide array of targets (*Teune et al., 2000*), we specifically examined whether cerebellar neurons projecting to ventral anterior/lateral (VAL) and central lateral (CL) thalamus differentially contribute to the rotarod learning and execution (*Figure 3*, detailed statistics in *Supplementary files 10–12*). For this purpose, an AAV5-hSyn-DIO-hM4D(Gi)-mCherry virus expressing an inhibitory DREADD conditioned to the presence of Cre-recombinase was injected into the CN, while a retrograde CAV-2 virus expressing the Cre recombinase was injected either in the CL or in the VAL. In both cases, we found an expression of hM4D(Gi)-mCherry throughout the CN mostly in the Interposed and Dentate for the CL injections (*Figure 3—figure supplement 1*). Since there was no effect of CNO in Sham mice (*Figure 1*), we only compared DREADD-injected animals receiving either CNO or SAL.

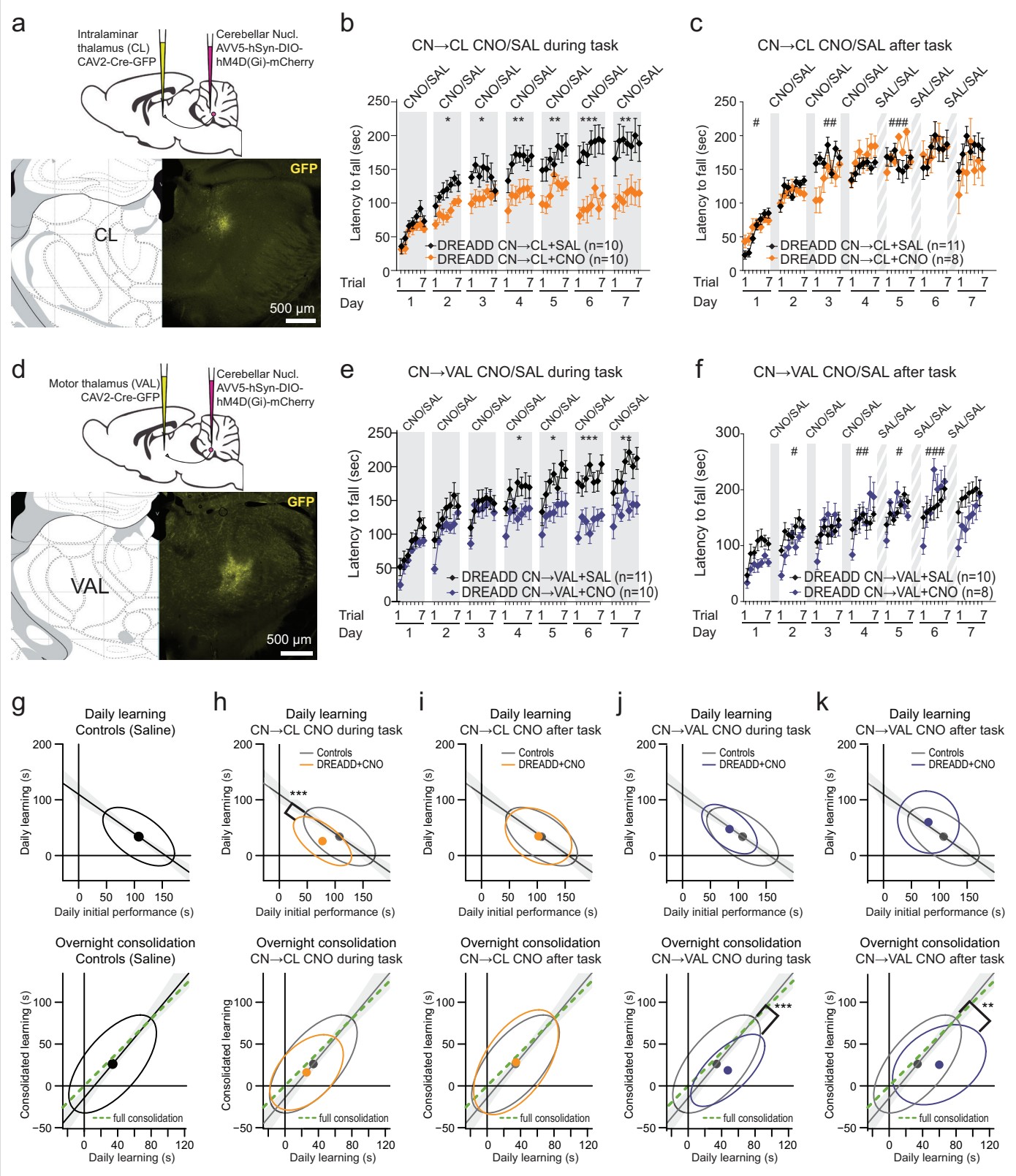

**Figure 3.** Inhibition of cerebellar nuclei (CN) neurons projecting to the centrolateral thalamus (CL) and to the ventral anterior lateral (VAL) thalamus during and after the training sessions differentially impairs motor learning. (**a**) Scheme of the combined viral injections targeting the CN->CL neurons using a retrograde virus expressing the Cre in the thalamus and a virus inducing Cre-dependent expression of inhibitory DREADD in the CN. *Top*: schematic of the viral injections. *Bottom*: GFP fluorescence revealing the site of injection of the CAV viruses. (**b**) Comparison of the effect of daily

*Figure 3 continued on next page*

*Figure 3 continued*

injections before trial 1 of CNO (orange) or Saline (black) in mice described in panel (**a**) (Data represents mean ± S.E.M, *n* indicates the number of mice; *p<0.05, **p<0.01, ***p<0.001; repeated measure ANOVA Group effect). (**c**) Same as (**b**) for daily CNO injections 30 min after trial 7 during the Early Phase. (**d**) Scheme of combined viral injections targeting the CN->VAL neurons. (**e,f**) Same as (**b,c**) for CNO-injected (blue) and control (black) mice obtained as described in panel d. (**g**) Same as *Figure 2c* for the Saline-treated groups of panels b–f. (**h–k**) Same as *Figure 2d–e* for the different groups receiving injections CNO during or after the task compared to Saline-treated mice. Same color coding as panels b, c and e, f. (**p<0.01 ***p<0.001 Wilcoxon test for difference between groups of residuals i.e. signed distance of performances to Deming regression line of control mice). CN: cerebellar nuclei, VAL: ventral anterior lateral thalamus, CL: central lateral thalamus.

The online version of this article includes the following figure supplement(s) for figure 3:

**Figure supplement 1.** Expression of hM4D(Gi)-mCherry in the cerebellar nuclei following CL and VAL injections.

**Figure supplement 2.** Inhibition of CN-CL or CN-VAL by 1 mg/kg CNO does not affect execution and fatigue, locomotion, motor coordination, balance, and strength.

**Figure supplement 3.** Retrograde infections from the CL and VAL differentially label cerebellar nuclei populations.

We examined the impact of the inhibition of cerebello-thalamic neurons on spontaneous locomotion, motor coordination, and strength. No significant differences were observed between the experimental groups for footprint patterns (*Figure 3—figure supplement 2*), grid test (*Figure 3—figure supplement 2B*), horizontal bar (*Figure 3—figure supplement 2C*), and vertical pole (*Figure 3—figure supplement 2D*), indicating that coordination and strength are not affected by the inhibition of cerebellar-thalamic pathways induced by the administration of CNO 1 mg/kg. We also determined locomotor activity in open-field experiments (*Figure 3—figure supplement 2E*, detailed statistics in *Supplementary file 14*), which revealed that velocity was generally not affected by CNO injection in DREADD or Sham mice (*Figure 3—figure supplement 2E*, detailed statistics in *Supplementary file 14*), although CNO-injected CN→VAL and CN→CL groups, respectively, exhibited slightly higher velocity on day 1 and lower velocity on day 4 in the first open-field session compared to the control group. No significant differences were observed between Saline vs. CNO-treated CN-CL mice in the fixed speed rotarod test (*Figure 3—figure supplement 2F*, detailed statistics in *Supplementary file 15*). However, we found a decrease in the latency to fall for 15 and 20 r.p.m. in the CN→VAL + CNO group, suggesting that the ability to locomote on a rotarod was slightly decreased, although the animals were still able to remain more than 1 min on the rotarod at 20 r.p.m. (*Figure 3—figure supplement 2F*, detailed statistics in *Supplementary file 15*).

We then examined how the accelerating rotarod learning was impaired by the inhibition of CN neurons involved in cerebello-thalamic pathways during (*Figure 3B and E*, detailed statistics in *Supplementary file 10*) and after (*Figure 3C and F*, detailed statistics in *Supplementary file 10*) the task. Inhibition of the CN→CL neurons during the task (*Figure 3B*, detailed statistics in *Supplementary file 10*) produced a progressive deviation from the performance of the control group during the Early Phase, yielding to a substantial reduction of performance in the Late phase. In contrast, when the inhibition took place after the task during days 1, 2, and 3 (*Figure 3C*, detailed statistics in *Supplementary file 10*), the performances remained similar to the control group. This suggests that the CN→CL neurons mostly contribute to the learning during training.

In contrast, the specific inhibition of the CN→VAL neurons yielded another pattern of performance evolution. First, when inhibition took place during the task (*Figure 3E*, detailed statistics in *Supplementary file 10*), there was a marked drop in performance from the last trial of one day to the first trial of the following day, as observed for the full DCN inhibition, and the performances saturated at a lower level than control mice during the Late phase. Second, when inhibition took place after the task on days 1, 2, and 3 (*Figure 3F*, detailed statistics in *Supplementary file 10*), a similar overnight drop in performance was found during the Early Phase. Interestingly, this drop was maintained when the treatment after the task was shifted from CNO to Saline after 4 days, suggesting the cerebellar contribution to the consolidation of the task is critical early in the learning process and cannot be easily reinstated later.

Then, we examined to which extent learning and consolidation were affected during the Early Phase by CN→CL versus CN→VAL neurons inhibition (*Figure 3G–K*, detailed statistics in *Supplementary file 11* and *Supplementary file 12*). Control mice also showed a pattern of decreased learning for higher initial performance and full overnight maintenance of the improvement of performance (*Figure 3G*, detailed statistics in *Supplementary file 11* and *Supplementary file 12*). Inhibition of the CN→CL

neurons reduced the learning compared to controls (i.e. smaller daily learning for similar daily initial performance) when performed during (*Figure 3H*) but not after (*Figure 3I*) the task and preserved learning consolidation (i.e. gain of performance is preserved overnight) in all cases (*Figure 3H and I*). In contrast, inhibition of the CN→VAL neurons during or after the task preserved the learning (i.e. same daily learning for similar initial performance) but selectively disrupted learning consolidation in all cases (*Figure 3J and K*). This indicates that CN→CL and CN→VAL neurons play different roles during the Early phase, the former primarily during the task, and the latter after the task.

The differential impact of chemogenetic inhibition of CN neurons retrogradely infected from the VAL and the CN suggests that different CN populations are targeted. To verify this, we performed combined retrograde injections of fluorescent rAAV centered on the VAL and CL (*Figure 3—figure supplement 3*). Retrograde injections yielded labeling in the three main divisions of the CN (Dentate, Interposed, and Fastigial). Retrogradely labeled cell soma from CL, VAL, or both were observed in each CN division and were distributed along anteroposterior axis (*Figure 3—figure supplement 3C*). Counts of labeled cells in each CN at 7 anteroposterior levels from each injection were highly correlated between the mice (*r*=0.92, 51 measures/mice) and counts were thus pooled. Overall, very few double-labeled neurons were found (3.4%, 30 double-labeled, 305 green, 549 red cells), consistent with a limited overlap of the CN populations retrogradely infected by the two types of injections.

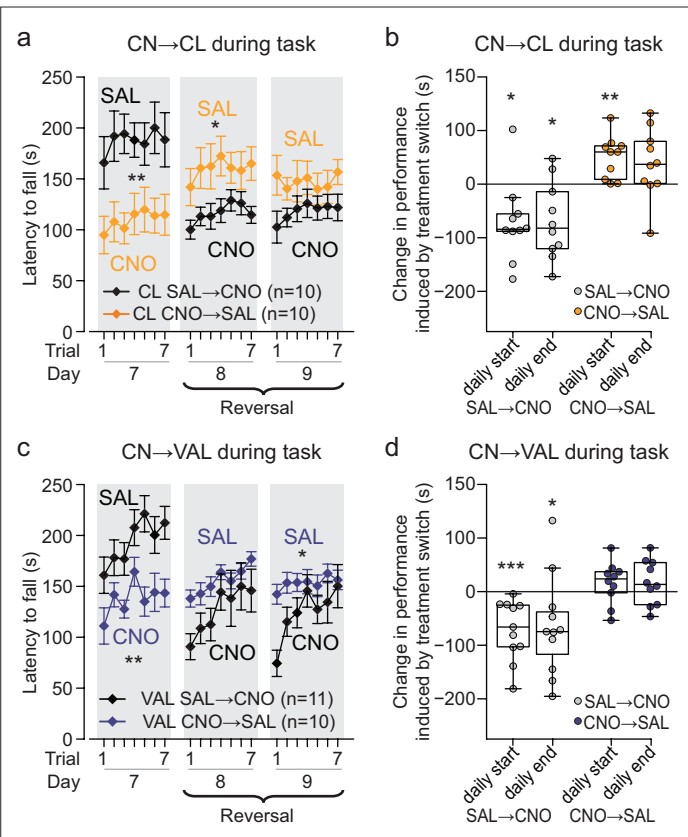

**Figure 4.** Reversal of the inhibition of the cerebello-thalamic neurons at the end of the Late phase yields lasting impairments. (**a**) Performance of mice with DREADD expression in CL-projecting CN neurons, which learned the task under respectively CNO and Saline treatment (*Figure 3a*) during 7 days and then received, respectively, Saline and CNO treatment during a Reversal phase (*: p<0.05 repeated-measure ANOVA group effect; Data represented as mean ± S.E.M, *n* indicates the number of mice). (**b**) Change induced by treatment switch on skill levels for the first trial (daily start) and last trial (daily end); values are estimated as in *Figure 2a*, and start and end values for days 8 and 9 are averaged for each animal; *p<0.05, **p<0.01, ***p<0.001 paired Wilcoxon test. Boxes represent quartiles and whiskers correspond to range; points are singled as outliers if they deviate more than 1.5 x interquartile range from the nearest quartile. (**c,d**) Same as panels a and b for mice with DREADD expression in VAL-projecting CN neurons (*Figure 3d*). CN: cerebellar nuclei, VAL: ventral anterior lateral thalamus, CL: central lateral thalamus.

## Inhibition of each type of cerebello-thalamic neurons impairs execution once maximal performance has been achieved

Mice that learned the task while either CN→CL or CN→VAL neurons were inhibited showed lower performance compared to controls after 7 days of training ('Day 7' in *Figures 3B and E and 4A and C*, detailed statistics in *Supplementary file 16*). To examine whether this result is due to the deficits in learning and consolidation described above or whether at this late stage, CN→CL or CN→VAL neurons participate in the task execution, we administered CNO to mice that had learned the accelerating rotarod during 7 days receiving Saline (*Figure 4*, black symbols). These CNO injections were thus performed at the end of the Late phase, when the performances of the mice were stable across days. In mice expressing inhibitory DREADD in CN→CL (*Figure 4A*, detailed statistics in *Supplementary file 16*) or CN→VAL neurons (*Figure 4C*, detailed statistics in *Supplementary file 16*), the injection of CNO before the task induced a significant reduction in performance both at the start and end of the daily training sessions (*Figure 4B and D*, detailed statistics in *Supplementary file 17*). Interestingly, daily learning and deficit in consolidation still took place under inhibition of CN→VAL neurons, while neither of these effects was present following inhibition of CN→CL neurons, consistent with the primary role of the latter for task learning and of the former for offline consolidation. These results, moreover, indicate that CN→CL and CN→VAL neurons contribute to task execution and/or memory retrieval in the late stage of learning in normal conditions.

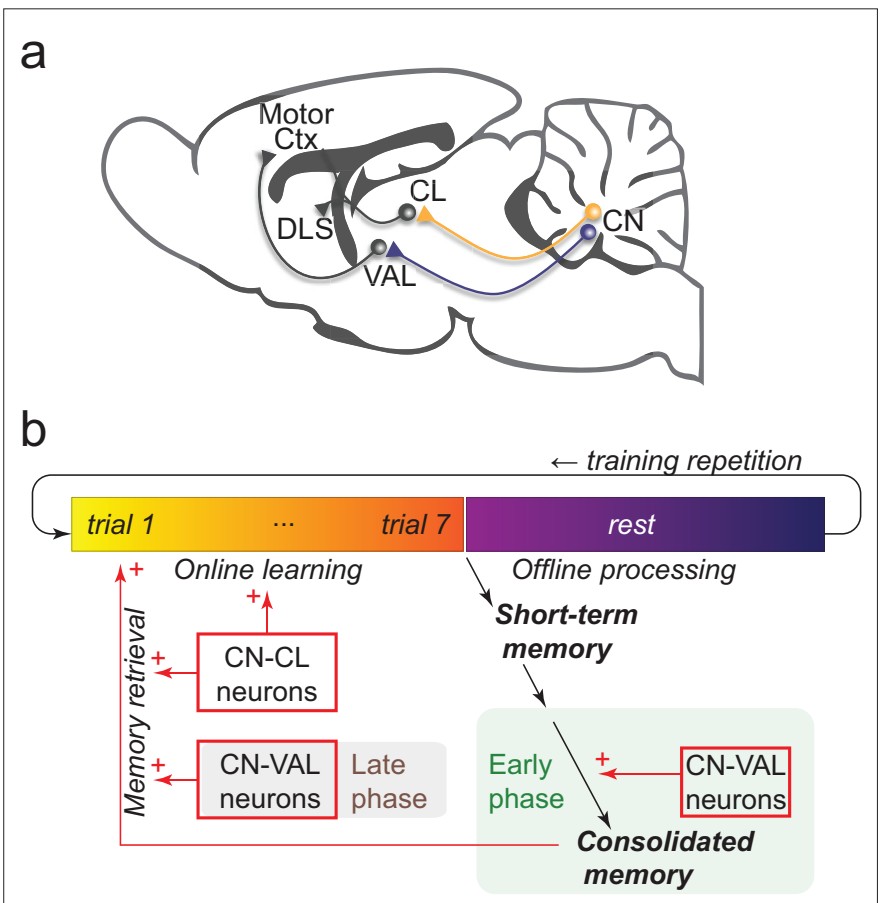

**Figure 5.** Summary of the behavioral findings. (**a**) Schematic representation of mouse brain. CN: Cerebellar Nuclei, CL: centrolateral thalamus, VAL: ventral and anterior thalamus, DLS: dorsolateral striatum, Motor Ctx: motor cortex. (**b**) Summary of the controls exerted by CN-CL and CN-VAL neurons on the skill learning. The CN-CL neurons contribute to online learning and retrieval/execution of the learned skill. The CN-VAL neurons contribute to consolidating offline the recent learning into a form of consolidated, readily-available, memory; a defect in consolidation in the early phases (first days) cannot be rescued in the late phase. CN-VAL neurons also participate in the retrieval/execution in the late phases of learning.

Since CN→CL or CN→VAL neurons participate in the performance in the Late phase, we expected that the removal of the inhibition after 7 days of training would result in an improvement of performance. Indeed, in mice expressing inhibitory DREADD in CN→CL neurons, there was a significant performance improvement (but no further learning) when the treatment was shifted from CNO to Saline administration (*Figure 4A and B*) consistent with the involvement of CN→CL neurons in task execution. Strikingly, such improvement was not observed when the inhibition of CN→VAL neurons was lifted (*Figure 4C and D*), indicating that the contribution of CN→VAL neurons in early phases of task learning is needed for a proper encoding of the task and that cannot be easily recovered if it was impaired at the early phases.

## Discussion

In this study, we found that a transient and mild chemogenetic inhibition which reduces the CN activity preserves the motor coordination but disrupts motor learning in a complex motor task, the accelerating rotarod task, known to depend on the motor cortex and basal ganglia. Moreover, we distinguish two contributions of the cerebellum to learning; one is carried by CN neurons projecting toward the intralaminar thalamus and is needed for learning and recall/execution. The other is carried by CN neurons projecting toward the motor thalamus and is required to perform an offline consolidation of a latent memory trace into a consolidated, readily available, motor skill (*Figure 5*). Finally, our results show that, beyond its role in learning and consolidation and independently from a role in basic motor coordination, the cerebellum becomes more strongly engaged via its projections toward the cerebral cortex and basal ganglia when the performance in the task progresses.

### A role for the cerebellum in the multi-nodal network of motor skill learning

By showing an involvement of the cerebellum in the accelerating rotarod task, our results complement the previous results which demonstrated that the basal ganglia and motor cortex are recruited and required to complete the task (*Costa et al., 2004*; *Cao et al., 2015*; *Kida et al., 2016*). Our chemogenetic experiments also indicate that the involvement of the cerebellum changes along the multiple phases of motor learning, ranging from a minimal contribution in the initial phase to a stronger contribution in the later phases. These observations parallel the converging evidence indicating that the areas of the basal ganglia involved in the accelerating rotarod evolve along the phases of learning (*Yin et al., 2009*; *Durieux et al., 2012*; *Cao et al., 2015*). Therefore, the three main central nodes of motor function (cortex, basal ganglia, and cerebellum) are differentially recruited along the multiple phases of the accelerating rotarod task.

The impact of various cerebellar disruptions on accelerating rotarod performance has previously been examined in too many studies to be listed exhaustively here, but the contribution of the cerebellum to learning is generally difficult to interpret. The reported effects range from ataxia and disruption of the ability to run on a rod (*Sausbier et al., 2004*), to normal learning (*Galliano et al., 2013*), defects in learning (*Groszer et al., 2008*; *Galliano et al., 2013*), defects in consolidation (*Sano et al., 2018*), and even increase in learning (*Iscru et al., 2009*). However, the cerebellum is critical for interlimb coordination (*Machado et al., 2015*; *Sathyamurthy et al., 2020*), and most studies lack proper motor controls to test the ability to walk on a rotating rod: poor performance may thus simply result from problems of running on the rod rather than problems of learning. Moreover, many studies involve genetic mutations which leave room for variable compensations along development and adult life, as exemplified by the diversity of the motor phenotypes of mice with different timing of degeneration of Purkinje cells (*Porras-García et al., 2013*). Finally, the multiphasic nature of rotarod learning is often overlooked.

Studies of cerebellar synaptic plasticity support the involvement of cerebellum in rotarod learning. Indeed, the targeted suppression of parallel fiber to Purkinje cell synaptic long-term depression in the cerebellar cortex disrupts rotarod learning after an initial phase, without altering any other motor ability tested (*Galliano et al., 2013*). Consistently, Thyrotropin-releasing hormone (TRH) knock-out mice do not express long-term depression at parallel fiber-Purkinje cell synapses and exhibit impaired performance in the late phase of rotarod learning, while the administration of TRH in the knock-out mice both restores long-term depression and accelerates rotarod learning (*Watanave et al., 2018*).

More generally, studies in mutant mice suggest that cerebellar plasticity is required for adapting skilled locomotion (*Vinueza Veloz et al., 2015*). This suggests that cerebellar plasticity is involved in accelerating rotarod learning and thus contributes to learning processes and not simply to the execution of the task. Yet it remains unclear whether this cerebellar learning is needed to improve descending control of the motor system, or whether it is needed by the motor centers in the forebrain.

## A specific impact on learning of CL-projecting CN neurons

In our study, we found that the chemogenetic inhibition of CN-CL neurons during the task reduces the learning performances of the mice. This effect likely results from basic motor deficits: we found that the chemogenetic modulation did not significantly alter (1) limb motor coordination in footprint analysis, (2) strength in the grid test, (3) speed in spontaneous locomotion in the open-field test, (4) locomotion speed and balance required to complete the horizontal bar test, and (5) body-limb coordination and balance required in the vertical pole test. Since all these motor parameters may be affected by cerebellar lesions, this suggests that CN-CL neurons are not necessary to maintain those functions, which might thus be relayed by other CN neurons; indeed, focal lesions in the intermediate cerebellum (thus projecting to the Interposed nuclei) have been reported to induce ataxia without altering rotarod learning (*Stroobants et al., 2013*). Alternatively, the effect of the partial inhibition induced by CNO in our study (typically ~35% reduction in firing rate) might be compensated at other levels in the motor system to ensure normal performances in these tasks; indeed, the selective ablation of CN-CL neurons has been reported to yield locomotor deficits in the initial performances on the accelerating rotarod (*Sakayori et al., 2019*), which contrasts with the lack of significant deficit in the early phase following CN-CL neurons inhibition in our study. A possible explanation for this discrepancy could be that our intervention, being milder than a full ablation, selectively disrupted the advanced patterns of locomotion only needed at the higher speeds of the rotarod and thus did not impact the slow rotarod locomotion typically performed in the initial learning phase of the task. However, the highest speeds reached on the rotarod correspond to the average locomotion speed in the open field, which is unaffected by the chemogenetic inhibition. Moreover, in our conditions, the inhibition of the CN-CL neurons did not produce significant deficits in the fixed-speed rotarod; CNO-treated animals ran on average for about 2 min at 20 r.p.m. while they fell on average after the same amount of time on the accelerating rotarod, corresponding to a rotarod speed below 20 r.p.m. at the time of the fall. This rules out a contribution of weakness, or fatigue, to the latency to fall in the accelerating rotarod, the CNO-treated animals being able to run on the fixed-speed rotarod more than twice the distance, at a higher speed, than the distance they run on the accelerating rotarod before falling. Overall, this indicates that the partial inhibition of the CN-CL neurons does not disrupt the elementary motor abilities needed in the task.

We observed that the daily gain in latency depended on the initial latency to fall of the day, both for control mice and for CN-CL inhibited mice. Therefore, we can compare learning intensity across conditions by examining how much learning takes place within a single day for a given initial performance. We found that inhibition of the CN-CL neurons yielded a lower increase in latency to fall for a given initial performance than in control mice, indicating a weaker learning. Moreover, the administration of CNO in animals that learned under Saline treatment after reaching maximal performance induced a sudden drop in performance, revealing a deficit in the execution of the learned task. Interestingly, animals that learned under CNO and were switched to Saline treatment after 7 days of treatment from the cerebello-thalamic inhibition showed little learning, suggesting that the cerebello-thalamic contribution is essential in the initial phase, possibly because of the encoding of the skill in different brain circuits once fully acquired (*Yin et al., 2009*). Overall, our results show that the CN-CL neurons contribute to both learning and execution of motor skills.

The inhibition of CN-VAL neurons during the task also yielded lower levels of performance in the Late stage, suggesting that these neurons contribute also to learning and/or execution of motor skills. Indeed, the mild defect in fixed speed rotarod could indicate the presence of a locomotor deficit, only visible at high speed. Although this locomotor deficit may not be critical in the first days (where the mice do not reach rotarod speeds of 20 rpm – i.e. 130 s) and where a learning deficit is already observed, the reduced maximal performance of mice following CN-VAL inhibition may thus in part result from motor deficit. Interestingly, both Dentate and Interposed nuclei contain some neurons with collaterals in both VAL and CL thalamic structures (*Aumann and Horne, 1996*; *Sakayori et al.,*

*2019*), suggesting that the effect on learning could be mediated by a combined action on the learning via the CL thalamus and via the VAL thalamus. However, consistent with *Sakayori et al., 2019*, we found that the manipulations of cerebellar neurons retrogradely targeted either from the CL or from the VAL produced different effects in the task. Indeed, our tracing experiments showed that retrograde infection from the CL and VAL labeled different populations. This indicates that either the distinct functional roles of VAL-projecting or CL-projecting neurons reported in our study are carried by a subset of pathway-specific neurons without collaterals, or that our retrograde infections in VAL and CL preferentially targeted different cerebello-thalamic populations even if these populations had axon terminals, probably of different densities, in both thalamic regions. Future studies mapping the different subpopulations of CN neurons using anterograde markers and single axon reconstruction shall help clarify this point.

Beyond the thalamus, the CL and VAL may project to overlapping targets (*Hunnicutt et al., 2014*), but the density of the projections may differ substantially. Since the CL is a narrow structure, accurate anterograde tracing is primarily available for the rat, and it shows only limited projections to the primary motor cortex, and the primary target seems to be the striatum and the cingulate and retrosplenial cortices (Figure 13 in *Van der Werf et al., 2002*), even if some retrograde labeling from the motor cortex in the CL has indeed been observed (*Cicirata et al., 1986*). Comparison of single axon reconstruction from the CL and VAL shows weaker branching patterns in the cortex for the CL (*Kuramoto et al., 2009*) vs (*Deschenes et al., 1996*). Given the scarce existing evidence for retrosplenial and cingulate cortices involvement in accelerating rotarod learning, an effect of the CL in learning through the dorsal striatum (*Costa et al., 2004*) seems more likely; however, the CL could also modulate learning through more minor cortical targets (secondary and primary motor cortices). Finally, the CL and VAL belong to different classes of thalamic profile (*Phillips et al., 2019*) consistent with their distinct connectivity and distinct functional role as found in our results.

## Contribution of VAL-projecting CN neurons to offline consolidation

While in control mice, the final performance at the end of a session could be reproduced at the beginning of the next session, this preservation of performance across night was heavily altered when CN-VAL neurons were inhibited after the task, suggesting an impairment of offline consolidation. However, in this group of mice, the daily gain of performance increased across days, instead of decreasing and compensating for the overnight loss. This faster relearning might reveal the presence of 'savings'. Therefore, if the inhibition of CN-VAL neurons alters the offline consolidation, a latent trace of the learning might remain, unaltered by CN-VAL inhibition, and allow for a faster relearning on the next day.

The effect of CNO peaks in less than an hour and lasts for several hours afterwards (*Alexander et al., 2009*); therefore, the disruption of offline consolidation reported above is produced by a disruption of the cerebellar activity in the few hours that follow the learning session. This falls in line with a number of evidence indicating that cerebellar-dependent learning is consolidated by the passage of time, even in the awake state (*Shadmehr and Holcomb, 1997*; *Muellbacher et al., 2002*; *Cohen et al., 2005*; *Doyon et al., 2009*; *Nagai et al., 2017*), although very few studies in humans have examined the impact of offline cerebellar stimulations on motor learning (*Samaei et al., 2017*). However, the recent observation of coordinated sleep spindles in the cortex and CN (*Xu et al., 2022*) provides a potential mechanism to a cerebellar involvement in consolidation since spindles are a cortical rhythm also associated with consolidation of motor learning (*Barakat et al., 2013*; *Lemke et al., 2021*).

In the case of rotarod, it has been noted that sleep is not required for the overnight preservation of performance (*Nagai et al., 2017*); however, sleep may still be required for the change of cortical (*Cao et al., 2015*; *Li et al., 2017*) or striatal neuronal substrate of the accelerating rotarod skill (*Yin et al., 2009*). Therefore, while the offline activity of CN could be more specifically associated in converting savings into readily available skills distributed over a wide circuit including the cortex and basal ganglia, multiple processes of memory consolidation would co-exist and operate at different timescales.

The existence of multiple timescales for consolidation has already been described in Human physiology where the movements could be consolidated without sleep while consolidation of goals (*Cohen et al., 2005*) or sequences (*Doyon et al., 2009*) would require sleep. It is indeed difficult, as for most real-life skills, to classify the accelerating rotarod as a pure locomotor adaptive learning, or a pure

locomotor sequence learning: on one hand, the shape of the rod and its rotation induce a change in the correspondence between steps and subsequent body posture and thus require some locomotor 'adaptation'. On the other hand, the acceleration of the rod introduces sequential aspects: (1) speed increases occur in a fixed sequence and (2) asymptotic performances require the use of successively multiple types of gaits as the trial progresses (*Buitrago et al., 2004*). Following offline inhibition of CN-VAL neurons, which aspect of the accelerating rotarod skill would be maintained and which would be lost? Faster relearning has been proposed to reflect an improved performance at selecting successful strategies (*Morehead et al., 2015*; *Ruitenberg et al., 2018*). An attractive possibility could be that novel sensory-motor correspondences encountered on the rotarod would remain learned, possibly leaving a memory trace within the cerebellum. Nevertheless, these elementary 'strategies' would not be properly temporally ordered into a sequence over the duration of a trial (~2 min) if the transfer to the cortex is disrupted. The learning session the next day would benefit from the existence of these fragments of skill (savings) in the cerebellum, but learning would still be required to order them properly. A similar idea has indeed been proposed for the contribution of the cerebellum to sequence learning (*Spencer and Ivry, 2009*). Alternatively, recent studies revealed cerebellar mechanisms which could serve sequence learning (*Ohmae and Medina, 2015*; *Khilkevich et al., 2018*) and the offline inhibition of CN-VAL neurons could disrupt the consolidation of these sequences via the feedback collaterals of CN-VAL neurons to the cerebellar cortex (*Houck and Person, 2015*). However, our study does not allow us to conclude on the nature of savings remaining after the offline inhibition of CN-VAL neurons.

Finally, our results on cerebellar consolidation of a task learning dependent on forebrain regions extend previous findings showing that simple oculomotor learning or adaptive reflexes, which are learned in the cerebellar cortex, undergo complete or partial consolidation via a transfer to CN or brainstem structures (*Bao et al., 2002*; *Kassardjian et al., 2005*; *Shutoh et al., 2006*). Our findings are also consistent with a dependence on post-learning neuronal activity (*Okamoto et al., 2011*). Moreover, these paradigms suggest that such consolidation processes may indeed support savings (*Medina et al., 2001*). While combined changes in the metabolic activity have been observed in the cerebellum and forebrain motor circuits along learning (e.g. *Shadmehr and Holcomb, 1997*; *Grafton et al., 2002*; *Della-Maggiore and McIntosh, 2005*; *Mawase et al., 2017*), such studies provide little information on cerebellar output since the metabolic activity mostly reflects input activity (*Howarth et al., 2012*). Our study thus complements such studies by providing support for an increasing role in cerebellar output neurons during the task along learning and consolidation.

In conclusion, our results provide clear evidence for the existence of online contributions of neurons belonging to distinct cerebello-thalamic pathways to the acquisition and execution of motor skills encoded in a cerebello-striato-cortical network. They also show a contribution to the offline consolidation by a distinct cerebello-thalamic population. Thus, our work highlights the importance of studying the contribution to learning of single nodes in the brain motor network from an integrated perspective (*Caligiore et al., 2017*; *Krakauer et al., 2019*) and supports a functional heterogeneity of cerebellar contributions to brain function (*Diedrichsen et al., 2019*).

## Materials and methods

**Key resources table**

| Reagent type (species) or resource | Designation | Source or reference | Identifiers | Additional information |
|---|---|---|---|---|
| Strain, strain background (*Mus musculus*, male) | C57BL/6 J | Charles River Laboratories | RRID:IMSR_JAX:000664 | Adult males, 8 weeks old |
| Recombinant DNA reagent | AAV5-hSyn-hM4D(Gi)-mCherry | University of North Carolina Vector Core | | Titer 7.4×10¹² vg/ml |
| Recombinant DNA reagent | AAV5-hSyn-EGFP | Penn Vector Core | | Control virus |
| Recombinant DNA reagent | AAV-hSyn-DIO-hM4D(Gi)-mCherry | University of North Carolina Vector Core | | Cre-dependent DREADD |
| Recombinant DNA reagent | CAV-2-Cre | Montpellier Vectorology Platform | | Cre-dependent DREADD |

*Continued on next page*

*Continued*

| Reagent type (species) or resource | Designation | Source or reference | Identifiers | Additional information |
|---|---|---|---|---|
| Chemical compound, drug | Clozapine-N-oxide (CNO) | Tocris Bioscience | Cat#:4936 | 1 mg/kg, i.p |
| Chemical compound, drug | Isoflurane | Baxter International | | Anesthesia |
| Chemical compound, drug | Paraformaldehyde | Sigma-Aldrich | | 4% fixation |
| Chemical compound, drug | Mowiol 4–88 | Sigma-Aldrich | | Mounting medium |
| Software, algorithm | Python | Python Software Foundation | RRID:SCR_008394 | Behavioral analysis |
| Software, algorithm | OpenCV | OpenCV | RRID:SCR_015799 | Video tracking |
| Software, algorithm | MATLAB | MathWorks | RRID:SCR_001622 | Data analysis |
| Software, algorithm | MountainSort v4 | Flatiron Institute | RRID:SCR_017675 | Spike sorting |
| Software, algorithm | SciPy | SciPy | RRID:SCR_008058 | Trajectory smoothing |
| Software, algorithm | R | The R Project for Statistical Computing | RRID:SCR_001905 | Statistical analysis |
| Software, algorithm | mcr package | The R Project for Statistical Computing | | Version 1.3.3.1 |
| Software, algorithm | ZEN Blue Edition | Carl Zeiss | | Image processing |
| Software, algorithm | Dvrtk | IGBMC | | Gait analysis |

## Animals

Adult male C57BL/6 J mice (Charles River, France, IMSR Cat# JAX:000664, RRID:IMSR_JAX:000664), 8 weeks of age and 24±0.4 g of weight at the beginning of the experiment were used in the study. Mice were fed with a chow diet and housed in a 22 °C animal facility with a 12 hr light/dark cycle (light phase 7 am–7 pm). The animals had free access to food and water. All animal procedures were performed in accordance with the recommendations contained in the European Community Directives (authorization number APAFIS#1334–2015070818367911 v3 and APAFIS #29793–202102121752192).

## Behavioral experiments

### Accelerating rotarod task

The rotarod apparatus (mouse rotarod, Ugo Basile) consisted of a plastic roller with small grooves running along its turning axis (*Bearzatto et al., 2005*). One week after injections, mice were trained with seven trials per day during 7 consecutive days. This training protocol was chosen since performance progression takes several days and reaches a plateau over a few days. During each trial, animals were placed on the rod rotating at a constant speed (4 r.p.m.), then the rod started to accelerate continuously from 4 to 40 r.p.m. over 300 s. The latency to fall off the rotarod was recorded. Animals that stayed on the rod for 300 s were removed from the rotarod and recorded as 300 s. Mice that clung to the rod for two complete revolutions were removed from the rod and time was recorded. Between each trial, mice were placed in their home cage for a 5 min interval.

### Open-field activity

Mice were placed in a circular arena made of plexiglass with 38 cm diameter and 15 cm height (Noldus, Netherlands) and video recorded from above. Each mouse was placed in the open field for a period of 10 min before and after the accelerating rotarod task with the experimenter out of its view. The position of the center of gravity of mice was tracked using an algorithm programmed in Python 3.5 and the OpenCV 4 library. Each frame obtained from the open-field videos was analyzed according to the following process: open-field area was selected and extracted in order to be transformed into a grayscale image. Then, a binary threshold was applied on this grayscale image to differentiate the mouse from the white background. To reduce the noise induced by the recording cable or by particles potentially present in the open field, a bilateral filter and a Gaussian blur were sequentially applied, since those components have a higher spatial frequency compared to the mouse. Finally, the

OpenCV implementation of the Canny algorithm was applied to detect the contours of the mouse, and the position of the mouse was computed as the mouse's center of mass. The trajectory of the center of mass was interpolated in x and y using scipy's Univariate Spline function (with smoothing factor s=0.2 x length of the data), allowing the extraction of a smoothed trajectory of the mouse. The distance traveled by the mouse between two consecutive frames was calculated as the variation of position of the mouse multiplied by a scale factor, to allow the conversion from pixel unit to centimeters. The total distance traveled was obtained by summing the previously calculated distances over the course of the entire open-field session. The speed was computed as the variation of position of center points on two consecutive frames divided by the time between these frames (the inverse of the number of frames per seconds). This speed was then averaged by creating sliding windows of 1 s. After each session, fecal boli were removed and the floor was wiped clean with a damp cloth and dried after the passing of each mouse.

### Horizontal bar

Motor coordination and balance were estimated with the balance beam test which consists of a linear horizontal bar extended between two supports (length: 90 cm, diameter: 1.5 cm, height: 40 cm from a padded surface). The mouse is placed in one of the sides of the bar and released when all four paws gripped it. The mouse must cross the bar from one side to the other, and latencies before falling are measured in a single trial session with a 3 min cut-off period.

### Vertical pole

Motor coordination was estimated with the vertical pole test. The vertical pole (51 cm in length and 1.5 cm in diameter) was wrapped with white masking tape to provide a firm grip. Mice were placed heads up near the top of the pole and released when all four paws gripped the pole. The bottom section of the pole was fixated to its home cage with the bedding present but without littermates. When placed on the pole, animals naturally tilt downward and climb down the length of the pole to reach their home cage. The time taken before going down to the home cage with all four paws was recorded. A 20 s habituation was performed before placing the mice at the top of the pole. The test was given in a single trial session with a 3 min cut-off period.

### Footprint patterns

Motor coordination was also evaluated by analyzing gait patterns using the approach used in *Simon et al., 2004*. Mouse footprints were used to estimate foot opening angle and hindbase width, which reflects the extent of muscle loosening. The mice crossed an illuminated alley, 70 cm in length, 8 cm in width, and 16 cm in height, before entering a dark box at the end. Their hindpaws were coated with nontoxic water-soluble ink and the alley floor was covered with sheets of white paper. To obtain clearly visible footprints, at least 3 trials were conducted. The footprints were then scanned and examined with the Dvrtk software (Jean-Luc Vonesch, IGBMC). The stride length was measured with hindbase width formed by the distance between the right and left hindpaws. Linearity, defined as the average change in angle between consecutive right-right steps, is calculated by drawing a line perpendicular to the direction of travel, starting at the first right footprint. After determining the angle between this perpendicular line and each subsequent right footprint, differences in angle were estimated between each consecutive step pair, and the average of absolute values of all angles was calculated. A high linearity score is indicative of nonlinear movement. Sigma, describing the regularity of step length, is defined as the standard deviation of all right-right and left-left step distance. Gait width, the average lateral distance between opposite left and right steps, is determined by measuring the perpendicular distance of a given step to a line connecting its opposite preceding and succeeding steps. Alternation coefficient, describing the uniformity of step alternation, is calculated by the mean of the absolute value of 0.5 minus the ratio of right-left distance to right-right step distance for every left-right step pair.

### Grid test

The grid test is performed to measure the strength of the animal. It consists of placing the animal on a grid which tilts from a horizontal position of 0° to 180°. The animal is registered by the side and the

time until it falls is measured. The time limit for this experiment is 30 s. In the cases where the mice climbed up to the top of the grid, a maximum latency of 30 s was applied.

### Fixed speed rotarod

Motor coordination, postural stability, and fatigue were estimated with the rotarod (mouse rotarod, Ugo Basile). Facing away from the experimenter's view, the mice placed on top of the plastic roller were tested at constant speeds (5, 10, 15, and 20 r.p.m). Latencies to fall were measured for up to 3 min in a single trial.

## Cerebellar outputs inactivation

We used evolved G-protein-coupled muscarinic receptors (hM4Di) that are selectively activated by the pharmacologically inert drug Clozapine-N-Oxide (CNO; *Alexander et al., 2009*). In our study, non-cre and cre-dependent versions of the hM4Di receptor packaged into an AAV were used in order to facilitate the stereotaxic-based delivery and regionally restricted the expression of hM4Di. As demonstrated previously (*Anaclet et al., 2014*; *Anaclet et al., 2015*; *Venner et al., 2016*; *Pedersen et al., 2017*; *Anaclet et al., 2018*), hM4Di receptor and ligand are biologically inert in the absence of ligand. Moreover, at the administered dose of 1 mg/kg, CNO injection induces a maximum effect during the 1–3 hr post-injection period (*Anaclet et al., 2014*; *Anaclet et al., 2018*), which enables us to confirm that during the whole duration of our protocols, the CNO was still effective. CNO administration in sham-operated animals and saline injection in sham-operated and DREADD-expressing animals were also tested to distinguish the effect of specific inhibition of the targeted neuronal population from a nonspecific effect of CNO or its metabolite clozapine (*Gomez et al., 2017*) or from the expression of DREADD without CNO.

In order to globally inactivate the cerebellar outputs, stereotaxic surgeries were used to inject DREADD viral constructs bilaterally into the Dentate, Interposed, and Fastigial nucleus. Mice were anesthetized with isoflurane for induction (3% in closed chamber during 4–5 min) and placed in the Kopf stereotaxic apparatus (model 942; PHYMEP, Paris, France) with mouse adapter (926-B, Kopf), and isoflurane vaporizer. Anesthesia was subsequently maintained at 1–2% isoflurane. A longitudinal skin incision was performed before removing the connective tissue on the skull and exposing the bregma and lambda sutures of the skull. The coordinates for the Dentate nucleus injections were: 6.2 mm posterior to bregma, +/-2.3 mm lateral to the midline and –2.4 mm from dura while the Interposed injections were placed anteroposterior (AP) –6.0 mm, mediolateral (ML) = +/-1.5 mm in respect to bregma and dorsoventral (DV) –2.1 mm depth from dura. Finally, the Fastigial injections were placed –6.0 AP,+/-0.75 ML in respect to bregma and –2.1 depth from dura. Small holes were drilled into the skull and DREADD (AAV5-hSyn-hM4D(Gi)-mCherry, University of North Carolina Viral Core, $7.4 \times 10^{12}$ vg per ml, 0.2 µl) or control (AAV5-hSyn-EGFP, UPenn Vector Core, the same concentration and amount) virus were delivered bilaterally via quartz micropipettes (QF 100-50-7.5,Sutter Instrument, Novato, USA) connected to an infusion pump (Legato 130 single syringe, 788130-KDS, KD Scientific, PHYMEP, Paris, France) at a speed of 100 nl/min. The micropipette was left in place for an additional 5 min to allow viral dispersion and prevent backflow of the viral solution into the injection syringe. The scalp wound was closed with surgical sutures, and the mouse was kept in a warm environment until resuming normal activity. All animals were given analgesic and fluids before and after the surgery.

## Chronic in vivo extracellular recordings in non-DREADD or DREADD mice

In a set of mice sham EGFP-injected or DREADD-injected mice (Dentate, Fastigial, and Interposed), bundles of electrodes were implanted into the CN. Both non-DREADD and DREADD injections (AAV5-hSyn-hM4D(Gi)-mCherry or AAV5-hSyn-EGFP) and electrodes implantation were performed the same day. This experiment was performed in order to evaluate and validate that hM4D(Gi) receptors decrease the activity within the three CN.

Recordings were performed in awake behaving control mice during the open-field sessions. Recordings and analysis were performed using an acquisition system with 32 channels (sampling rate 25 kHz; Tucker Davis Technology System 3) as described in *de Solages et al., 2008*; *Popa et al., 2013*. Bundles of electrodes consisting of nichrome wire (0.005 inches diameter, Kanthal RO-800)

folded and twisted into six to eight channels were implanted (electrode tip located at Fastigial: –6.0 AP,+/-0.75 ML, –2.1 depth from dura; Interposed: –6.0 AP, +/-1.5 ML, –2.1 depth from dura; Dentate: –6.2 AP, +/-2.3 ML, –2.4 depth from dura). To protect these bundles and ensure a good electrode placement, they were then held through a metal tube (8–10 mm length, 0.16–0.18 mm inner diameter, Coopers Needle Works Limited, UK) attached to an electrode interface board (EIB-16 or EIB-32; Neuralynx) by Loctite universal glue. Microwires of each bundle were connected to the EIB with gold pins (Neuralynx). The entire EIB and its connections were secured in place by dental cement for protection purposes. Electrodes were cut to the desired length before implantation (extending 0.5 mm below tube tip). The 1 kHz impedance of each electrode was measured and lowered by gold-plating to 200–500 kΩ. Mice were anesthetized with isoflurane and placed in the stereotaxic apparatus, then the skull was drilled and dura were removed above CN recording site (see Cerebellar outputs inactivation for a detailed description of the surgical procedure). Electrode bundles were lowered into the brain, the ground was placed above the cerebellar cortex, and the assembly was secured with dental cement. One week after the surgery to allow for virus expression, we started to record cellular activity in the CN in freely moving mice placed in the open field. Mice were habituated to the recording cable for 2–3 days before starting the recording. Recordings in the open field were performed before and after CNO or saline (SAL) injection. The mice were recorded for a 10 min baseline period followed by intraperitoneal injections of CNO 1 mg/kg or SAL, which were performed in a random sequence using a crossover design. After CNO or SAL injection, the mice were recorded during 30 min before and 15 min after the accelerating rotarod task protocol. Signal was acquired by headstage and amplifier from TDT (RZ2, RV2, Tucker-Davis Technologies, USA) and analyzed with Matlab and Python 3.5. The mice were not recorded until CNO washout, but successive days yielded similar starting firing rates.

The spike sorting was performed with MountainSort version 4 (*Chung et al., 2017*; https://github.com/flatironinstitute/mountainsort; *Magland, 2023*). Single units were isolated based on quantitative quality metrics computed after MountainSort 4 spike sorting (Python 3.8). Units were required to have a signal-to-noise ratio (SNR) greater than 5, inter-spike interval (ISI) violations less than 1%, an amplitude cutoff below 0.1, a presence ratio above 0.9, a firing rate greater than 0.1 Hz, and at least 50 detected spikes. In addition, units were assessed for temporal stability across the recording using autocorrelograms and presence over the recording, ensuring there were no prolonged periods of total inactivity. Units meeting these criteria were deemed well-separated and reliable for further analysis (*Figure 1—figure supplement 3*).

The average firing rates were computed from the recordings during the open-field sessions. At the end of experiments, the placement of the electrodes was verified. Firing rate modulation was computed as the difference of firing rate after and before CNO injection divided by the average of these two firing rates; it is therefore bounded between –100% (total suppression) and +100% (discharge only post-treatment).

## Cerebellar-thalamic outputs inactivation

In order to inhibit specifically cerebellar outputs to the centrolateral (CL) and/or ventral anterior lateral (VAL) thalamus, we applied a pathway-specific approach (*Boender et al., 2014*). The technique comprises the combined use of a CRE-recombinase expressing canine adenovirus-2 (CAV-2) injected in the thalamus and an adeno-associated virus (AAV-hSyn-DIO-hM4D(Gi)-mCherry) that contains the floxed inverted sequence of the DREADD hM4D(Gi)-mCherry injected in the CN. It entails the infusion of these two viral vectors into two sites that are connected through direct neuronal projections. AAV-hSyn-DIO-hM4D(Gi)-mCherry is infused in the site where the cell bodies are located, while CAV-2 is infused in the area that is innervated by the corresponding axons. After infection of axonal terminals, CAV-2 is transported towards the cell bodies and expresses CRE-recombinase (*Kremer et al., 2000*; *Hnasko et al., 2006*). AAV-hSyn-DIO-hM4D(Gi)-mCherry contains the floxed inverted sequence of hM4D(Gi)-mCherry, which is reoriented in the presence of CRE, prompting the expression of hM4D(Gi)-mCherry. This ensures that hM4D(Gi)-mCherry is not expressed in all AAV-hSyn-DIO-hM4D(Gi)-mCherry infected neurons, but exclusively in those that are also infected with CAV-2. Using the same procedures described above, 0.4 μl of the retrograde canine adeno-associated cre virus (CAV-2-cre, titter ≥ 2.5 × 10⁸) (Plateforme de Vectorologie de Montpellier, Montpellier, France) was bilaterally injected in the CL (from bregma: AP –1.70 mm, ML ± 0.75 mm, DV –3.0 mm) and VAL (from bregma: AP –1.4 mm, ML ± 1.0 mm, DV –3.5 mm). In addition, 0.2 μl of AAV-hSyn-DIO-hM4D(Gi)-mCherry

(UNC Vector Core, Chapel Hill, NC, USA) was bilaterally injected 1 week later into the CN, focusing on the Dentate (from bregma: AP −6.2 mm, ML ± 2.3 mm, DV −2.4 mm) and Interposed (from bregma: AP −6.0 mm, ML ± 1.5 mm, DV −2.1 mm) nucleus. All the stereotactic coordinates were determined based on The Mouse Brain Atlas (*Paxinos and Franklin, 2004*).

## Behavioral experiments design

Behavioral tests were performed one week following stereotaxic surgery to allow for virus expression. Balance beam, vertical pole, footprint patterns, grid test, and fixed speed rotarod experiments were performed 30 min after CNO (1 mg/kg, ip) or SAL injections. Two different strategies were used for the accelerating rotarod motor learning task experiments: (1) CNO (1 mg/kg, ip) or SAL was injected every day 30 min before the 1st trial of the accelerating rotarod task. Four days later, to ensure a proper CNO washout, mice were retested by receiving 7 trials for two consecutive daily sessions. Drug-free mice received CNO (1 mg/kg) or SAL 30 min before the first trial in both days. The treatments were inverted, meaning that those animals that received CNO during the preceding 7 days in this case were injected with SAL and the other way around. (2) CNO (1 mg/kg, ip) was injected 30 min after last trial at the days 1, 2, and 3; subsequently, mice received SAL 30 min after last trial at the days 4, 5, and 6 of the accelerating rotarod task.

The DREADD ligand Clozapine-N-Oxide (CNO, TOCRIS, Bristol, UK) was dissolved in SAL (0.9% sodium chloride) and injected intraperitoneally at 1 mg/kg.

## Histology

Mice were anesthetized with ketamine/xylazine (100 and 10 mg/kg, i.p., respectively) and rapidly perfused with ice-cold 4% paraformaldehyde in phosphate buffered SAL (PBS). The brains were carefully removed, postfixed in 4% paraformaldehyde for 24 hr at 4 °C, cryoprotected in 20% sucrose in PBS. The whole brain was cut into 40-µm-thick coronal sections on a cryostat (Thermo Fisher Scientific HM 560; Waltham, MA, USA). The sections were mounted on glass slides sealed with Mowiol mounting medium (Mowiol 4–88; Sigma-Aldrich, France). Verification of virus injection site and DREADDs expression was assessed using a wide-field epifluorescence microscope (BX-43, Olympus, Waltham, MA, USA) using a mouse stereotaxic atlas (*Paxinos and Franklin, 2004*). We only kept mice showing a well-targeted viral expression centered on the targeted nucleus. Representative images of virus expression were acquired on a Zeiss 800 Laser Scanning Confocal Microscope (×20 objective, NA 0.8; Carl Zeiss, Jena, Germany). Images were cropped and annotated using Zeiss Zen 2 Blue Edition software.

## Quantification and statistical analysis

Latency to fall (mean ± S.E.M) in rotarod for chemogenetic experiments was analyzed using one-way ANOVA repeated measure followed by two types of Posthoc tests: paired t-test for repeated-measure comparison and independent t-test for cross-group comparisons. Locomotor activity (velocity) in open-field (mean ± S.E.M) was analyzed using two-way ANOVA repeated measure (treatment × moment) followed by t-test Posthoc (comparisons between treatments for each open-field session). Fixed speed rotarod (mean ± S.E.M) was analyzed using two-way ANOVA repeated measures (treatment × speed) followed by t-test Posthoc (comparisons between treatments for each speed step). Footprint patterns parameters, horizontal bar, vertical pole, and grid test were analyzed using one-way ANOVA. Data are represented as boxplots (median quartiles and interquartile range plus outliers).

Latency to fall exhibits variations between successive trials, so that single trial performances are poor estimators of the skill level. To get a more robust estimate of the initial and final skill level and thus of learning of each day, we performed a linear regression on the latency to fall for each day and each animal; the within-day learning and overnight loss was estimated from the start- and end-points (corresponding to trial 1 and 7) of each regression segment (*Figure 2a*). To estimate the interdependence of initial performance of the day, within-day learning, and inter-day learning, we used Deming linear regression, assuming equal variance of the random noise of the measured quantity on the x- and y-axes. This assumption is made because the measures used in the Deming regressions are all derived from latency to fall, which is modeled as a combination of skill level and random noise. Deming confidence intervals were obtained by bootstrap. These values were computed with the R package mcr (version 1.3.3.1). To test if treatments altered the relationship between *initial performance* vs *learning*

or *daily learning* vs *overnight learning*, we compared the distribution of signed distance ('residuals') to the control Deming regression line between groups. Indeed, while in control groups these quantities show clear linear relationships, this is far less the case in treatment groups (possibly due to the variability of the effect of the treatment - efficacy of viral injections - and/or to the disruption of the neurobiological processes underlying these relationships) and thus precludes direct comparisons of the regressions between groups. Therefore, to test if there is a change in these relationships following treatments, we examined to which extent data points from treatment groups in bivariate comparisons (*initial perf * daily learning*, *daily learning x consolidated learning*) are distributed around the control group regression line differently from the data points of control groups. We therefore use a Wilcoxon comparison of the distributions of residuals of the treatment groups vs residuals of control groups around the control regression line to test if the relationship examined is disrupted by the treatment for this group: for example if the median of residuals of the treatment group is significantly lower than the median of the control group in the *initial performance * daily learning* comparison, it indicates that learning is slower, taking into account its dependence on initial performance. Similarly, if the residuals of the treatment group are lower than those of the control group in the *daily learning * consolidated learning* comparison, it indicates that consolidation is weaker.

## Acknowledgements

The authors thank David Robbe, Philippe Isope, and Sang-Jeong Kim for critical reading of the manuscript. This work was supported by Agence Nationale de Recherche to DP (ANR-19-CE37-0007-01 Multimod, ANR-21-CE37-0025 CerebellEMO, ANR-24-CE37-3944 Ce-Multi-TimeS) and to CL (ANR-17-CE16-0019 Synpredict, ANR-21-CE16-0017 PomPom), by Fondation pour la Recherche Médicale FRM EQU-202103012770 to CL and by the Institut National de la Santé et de la Recherche Médicale (France). We gratefully acknowledge the IBENS imaging facility (IMACHEM-IBiSA), member of the French National Research Infrastructure France-BioImaging (ANR-10-INBS-04), which received support from the "Fédération pour la Recherche sur le Cerveau - Rotary International France" (2011) and from the program « Investissements d'Avenir » ANR-10-LABX-54 MEMOLIFE. This work has received support under the Major Research Program of PSL Research University "PSL-NEURO", launched by PSL Research University and implemented by ANR (ANR-10-IDEX-0001).

## Additional information

### Funding

| Funder | Grant reference number | Author |
| --- | --- | --- |
| Agence Nationale de la Recherche | ANR-19-CE37-0007-01 | Daniela Popa |
| Agence Nationale de la Recherche | ANR-21-CE37-0025 | Daniela Popa |
| Agence Nationale de la Recherche | ANR-17-CE16-0019 | Clément Léna |
| Agence Nationale de la Recherche | ANR-21-CE16-0017 | Clément Léna |
| Fondation pour la Recherche Médicale | EQU-202103012770 | Clément Léna |
| Agence Nationale de la Recherche | ANR-24-CE37-3944 | Daniela Popa |
| Agence Nationale de la Recherche | ANR-10-IDEX-0001 PSL-NEURO | Clément Léna Daniela Popa |
| Agence Nationale de la Recherche | ANR-10-INBS-04 IMACHEM-IBiSA | Clément Léna Daniela Popa |

| Funder | Grant reference number | Author |
| --- | --- | --- |

The funders had no role in study design, data collection and interpretation, or the decision to submit the work for publication.

## Author contributions

Andrés Pablo Varani, Conceptualization, Data curation, Formal analysis, Supervision, Investigation, Methodology, Writing – original draft; Caroline Mailhes-Hamon, Investigation, Methodology; Romain W Sala, Data curation, Formal analysis, Investigation; Marie Sarraudy, Jimena L Frontera, Data curation, Investigation; Sarah Fouda, Data curation, Formal analysis; Clément Léna, Data curation, Software, Formal analysis, Funding acquisition, Validation, Investigation, Visualization, Methodology, Writing – original draft, Writing – review and editing; Daniela Popa, Conceptualization, Formal analysis, Supervision, Funding acquisition, Validation, Investigation, Visualization, Methodology, Writing – original draft, Project administration, Writing – review and editing

## Author ORCIDs

Andrés Pablo Varani ![ORCID] https://orcid.org/0000-0003-4227-5785
Caroline Mailhes-Hamon ![ORCID] https://orcid.org/0000-0002-7009-7798
Romain W Sala ![ORCID] https://orcid.org/0000-0002-6768-043X
Marie Sarraudy ![ORCID] https://orcid.org/0009-0004-4546-1203
Sarah Fouda ![ORCID] https://orcid.org/0009-0007-1844-5373
Clément Léna ![ORCID] https://orcid.org/0000-0002-1431-7717
Daniela Popa ![ORCID] https://orcid.org/0000-0002-8389-1122

## Ethics

All animal procedures were performed in accordance with the recommendations contained in the European Community Directives (authorization number APAFIS#1334-2015070818367911 v3 and APAFIS #29793-202102121752192).

Reviewer #2 (Public review): https://doi.org/10.7554/eLife.102813.3.sa1
Reviewer #3 (Public review): https://doi.org/10.7554/eLife.102813.3.sa2
Author response https://doi.org/10.7554/eLife.102813.3.sa3

# Additional files

## Supplementary files

Supplementary file 1. Statistics for *Figure 1g*.

Supplementary file 2. Statistics for *Figure 1h*.

Supplementary file 3. Statistics for *Figure 1i and j* part 1.

Supplementary file 4. Statistics for *Figure 1i and j* part 2.

Supplementary file 5. Statistics for *Figure 1j*.

Supplementary file 6. Statistics for *Figure 1—figure supplement 2a and b*.

Supplementary file 7. Statistics for *Figure 1—figure supplement 2c*.

Supplementary file 8. Statistics for *Figure 2c* controls.

Supplementary file 9. Statistics for *Figure 2de*. Deming regression.

Supplementary file 10. Statistics for *Figure 3b, c, e and f*.

Supplementary file 11. Statistics for *Figure 3g* Controls.

Supplementary file 12. Statistics for *Figure 3g*. Deming regression.

Supplementary file 13. Statistics for *Figure 3—figure supplement 2a-d*.

Supplementary file 14. Statistics for *Figure 3—figure supplement 2e*.

Supplementary file 15. Statistics for *Figure 3—figure supplement 2f*.

Supplementary file 16. Statistics for *Figure 4a and c*.

Supplementary file 17. Statistics for *Figure 4b and d*.

MDAR checklist

## Data availability

The data is available on a Dryad repository: https://doi.org/10.5061/dryad.51c59zwpw.

The following dataset was generated:

| Author(s) | Year | Dataset title | Dataset URL | Database and Identifier |
|---|---|---|---|---|
| Varani A, Léna C, Popa D | 2026 | Latency to fall in rotarod task for mice tested with various inhibition of cerebellar nuclei neurons and corresponding control groups | https://doi.org/10.5061/dryad.51c59zwpw | Dryad Digital Repository, 10.5061/dryad.51c59zwpw |

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
