## [Editor Report · eLife Assessment]

Varani et al present **important** findings regarding the role of distinct cerebellothalamic connections in motor learning and performance. The evidence supporting the main claims is **convincing**, with multiple replications, validation of their techniques, and appropriate controls. The work will be of broad interest to neuroscientists interested in central mechanisms of motor learning and control, as well as thalamic physiology.

---

## [Referee Report · Reviewer #2 (Public review)]

Summary:

This study examines the contribution of cerebello-thalamic pathways to motor skill learning and consolidation in an accelerating rotarod task. The authors use chemogenetic silencing to manipulate activity of cerebellar nuclei neurons projecting to two thalamic subregions that target motor cortex and striatum. By silencing these pathways during different phases of task acquisition (during task vs after task), the authors report valuable finding of the involvement of these cerebellar pathways in learning and consolidation.

Strengths:

The experiments are well-executed. The authors perform multiple controls and careful analysis to solidly rule out any gross motor deficits caused by their cerebellar nuclei manipulation. The finding that cerebellar projections to the thalamus are required for learning and execution of the accelerating rotarod task adds to a growing body of literature on the interactions between the cerebellum, motor cortex, and basal ganglia during motor learning. The finding that silencing the cerebellar nuclei after task impairs consolidation of the learned skill is interesting.

Revision comment:

The revised manuscript is improved in clarity and methodological detail. An important addition is the retrograde labeling data showing a degree of anatomical segregation between CN->CL and CN->VAL pathways that strengthens their reported different functional roles. I still think that potential effects on motor execution when cerebellar nuclei are silenced during task performance may complicate interpretations specifically related to learning. However, the evidence supporting a role of the cerebellar nuclei in off-line consolidation is convincing.

Overall, the study outlines a multifaceted role of the cerebellum in motor learning, consolidation, and execution. The demonstration that cerebellar projections to distinct forebrain structures contribute to these processes is significant.

---

## [Referee Report · Reviewer #3 (Public review)]

Summary:

Varani et al present important findings regarding the role of distinct cerebellothalamic connections in motor learning and performance. Their key findings are that: (1) cerebellothalamic connections are important for learning motor skills, (2) cerebellar efferents specifically to the central lateral (CL) thalamus are important for short-term learning, (3) cerebellar efferents specifically to the ventral anterior lateral (VAL) complex are important for offline consolidation of learned skills, and (4) that once a skill is acquired, cerebellothalamic connections become important for online task performance. The authors went to great lengths to separate effects on motor performance from learning, for the most part successfully. While one could argue about some of the specifics, there is little doubt that the CN-CL and CN-VAL pathways play distinct roles in motor learning and performance. An important next step will be to dissect the downstream mechanisms by which these cerebellothalamic pathways mediate motor learning and adaptation.

Strengths:

(1) The dissociation between on-line learning through CN-CL and offline consolidation through CN-VAL is convincing.

(2) The ability to tease learning apart from performance using their titrated chemogenetic approach is impressive. In particular, their use of multiple motor assays to demonstrate preserved motor function and balance is an important control.

(3) The evidence supporting the main claims is convincing, with multiple replications of the findings and appropriate controls.

(4) The retrograde tracing experiments (Supplementary Figure 5) demonstrate convincingly that the CN-VAL and CN-CL projections are almost entirely segregated,

Weaknesses:

(1) Despite the care the authors took to demonstrate that their chemogenetic approach does not impair online performance, there is (as they acknowledge in the Discussion) impaired rotarod performance at fixed higher speeds in Supplementary Figure 4f for CN-VAL projections, suggesting that there could be subtle changes in motor performance below the level of detection of their assays. There is also a trend in the same direction that did not pass significance for CN-CL at higher speeds, suggesting that part of the effects could be related to subtle deficits in performance.

---

## [Author Response]

The following is the authors’ response to the original reviews.

**eLife Assessment**
This study provides evidence that cerebellar projections to the thalamus are required for learning and execution of motor skills in the accelerating rotarod task. This important study adds to a growing body of literature on the interactions between the cerebellum, motor cortex, and basal ganglia during motor learning. The data presentation is generally sound, especially the main observations, with some limitations in describing the statistical methods and a lack of support for two separate cerebello-thalamic pathways, which is incomplete in supporting the overall claim.

We completed the MS by adding a double retrograde labelling study showing that the two pathways have limited overlap and by addressing the other concerns.

**Public Reviews:**

**Reviewer #1 (Public review):**
This is an interesting manuscript tackling the issue of whether subcircuits of the cerebellum are differentially involved in processes of motor performance, learning, or learning consolidation. The authors focus on cerebellar outputs to the ventrolateral thalamus (VL) and to the centrolateral thalamus (CL), since these thalamic nuclei project to the motor cortex and striatum respectively, and thus might be expected to participate in diverse components of motor control and learning. In mice challenged with an accelerating rotarod, the investigators reduce cerebellar output either broadly, or in projection-specific populations, with CNO targeting DREADD-expressing neurons. They first establish that there are not major control deficits with the treatment regime, finding no differences in basic locomotor behavior, grid test, and fixed-speed rotarod. This is interpreted to allow them to differentiate control from learning, and their inter-relationships. These manipulations are coupled with chronic electrophysiological recordings targeted to the cerebellar nuclei (CN) to control for the efficacy of the CNO manipulation. I found the manuscript intriguing, offering much food for thought, and am confident that it will influence further work on motor learning consolidation. The issue of motor consolidation supported by the cerebellum is timely and interesting, and the claims are novel. There are some limitations to the data presentation and claims, highlighted below, which, if amended, would improve the manuscript.

We thank the reviewer for the positive comments and insightful critics.

(1) Statistical analyses: There is too little information provided about how the Deming regressions, mean points, slopes, and intercepts were compared across conditions. This is important since in the heart of the study when the effects of inactivating CL- vs VL- projecting neurons are being compared to control performance, these statistical methods become paramount. Details of these comparisons and their assumptions should be added to the Methods section. As it stands I barely see information about these tests, and only in the figure legends. I would also like the authors to describe whether there is a criterion for significance in a given correlation to be then compared to another. If I have a weak correlation for a regression model that is non-significant, I would not want to 'compare' that regression to another one since it is already a weak model. The authors should comment on the inclusion criteria for using statistics on regression models.

We thank the reviewer for pointing out this weakness of description. The description of the Methods has thus been expanded and better justified in the “Quantification and statistical analysis” section.

We agree with the reviewer that comparison between Deming regressions would be fragile due to the weakness of these regression in treatment groups (while they are quite robust for control groups) and they are not included in the MS, although Deming regression coefficients with their confidence intervals are now provided for all groups in the statistical tables. As now more clearly explained in the Methods, the comparisons between groups are based on the distribution of residuals around regressions of the control regression lines. If we understand correctly the reviewer’s request, the control groups are all included.

(2) The introduction makes the claim that the cerebellar feedback to the forebrain and cortex are functionally segregated. I interpreted this to mean that the cerebellar output neurons are known to project to either VL or CL exclusively (i.e. they do not collateralize). I was unaware of this knowledge and could find no support for the claim in the references provided (Proville 2014; Hintzer 2018; Bosan 2013). Either I am confused as to the authors' meaning or the claim is inaccurate. This point is broader however than some confusion about citation.

The references are not cited in the context of collaterals from the DCN but for the output channels of the basal ganglia and cerebellum: “They [basal ganglia and cerebellum] send projections back to the cortex via anatomically and functionally segregated channels, which are relayed by predominantly non-overlapping thalamic regions (Bostan, Dum et al. 2013, Proville, Spolidoro et al. 2014, Hintzen, Pelzer et al. 2018).” Indeed, the thalamic compartments targeted by the basal ganglia and cerebellum are distinct, and in the Proville 2014, we showed some functional segregation of the cerebello-cortical projections (whisker vs orofacial ascending projections). Hintzen et al. have indeed performed an extensive review indicating the limited overlap between cerebellar- and basal ganglia-recipient territories. The sentence has been corrected to clarify what the “They” referred to.

The study assumes that the CN-CL population and CN-VL population are distinct cells, but to my knowledge, this has not been established. It is difficult to make sense of the data if they are entirely the same populations, unless projection topography differs, but in any event, it is critical to clarify this point: are these different cell types from the nuclei? how has that been rigorously established?; is there overlap? No overlap? Etc. Results should be interpreted in light of the level of this knowledge of the anatomy in the mouse or rat.

There is indeed a paragraph devoted to the discussion of this point (last part of the section “A specific impact on learning of CL-projecting CN neurons.”). Briefly, we actually know from the literature that there is a degree of collateralization (CN neurons projecting to both VAL and CL, see refs cited above), but as the reviewer says, it does not seem logically possible that the exact same population would have different effects, which are very distinct during the first learning days. The only possible explanation is the CN-CL and CN-VAL infections recruit somewhat different populations of neurons. We have now added more experiments to support our finding using retrograde infections using two rAAV viruses expressing red and green fluorescent reporter. These experiments confirm the limited overlap of the two populations of interest obtained by retrograde infection. We feel thus confident that while some CN neurons may project to both structures, retrograde infection strategies thus appear to differentially infect CN populations.

(3) It is commendable that the authors perform electrophysiology to validate DREADD/CNO. So many investigators don't bother and I really appreciate these data. Would the authors please show the 'wash' in Figure 1a, so that we can see the recovery of the spiking hash after CNO is cleared from the system? This would provide confidence that the signal is not disappearing for reasons of electrode instability or tissue damage/ other.

The recordings were not extended to the wash period, but examination of the firing rate before CNO on successive days did not evidence major changes in the population firing rate (this is now shown in a new supplementary figure 6).

(4) I don't think that the "Learning" and "Maintenance" terminology is very helpful and in fact may sow confusion. I would recommend that the authors use a day range " Days 1-3 vs 4-7" or similar, to refer to these epochs. The terminology chosen begs for careful validation, definitions, etc, and seems like it is unlikely uniform across all animals, thus it seems more appropriate to just report it straight, defining the epochs by day. Such original terminology could still be used in the Discussion, with appropriate caveats.

Since reference to these time windows is repeatedly used in the text we have shifted to “Early” and “Late” phase terminology.

(5) Minor, but, on the top of page 14 in the Results, the text states, "Suggesting the presence of a 'critical period' in the consolidation of the task." I think this is a non-standard use of 'critical period' and should be removed. If kept, the authors must define what they mean specifically and provide sufficient additional analyses to support the idea. As it stands, the point will sow confusion.

This has been corrected to: “suggesting the cerebellar contribution to the consolidation of the task is critical early in the learning process and cannot be easily reinstated later”

**Reviewer #2 (Public review):**
Summary:This study examines the contribution of cerebello-thalamic pathways to motor skill learning and consolidation in an accelerating rotarod task. The authors use chemogenetic silencing to manipulate the activity of cerebellar nuclei neurons projecting to two thalamic subregions that target the motor cortex and striatum. By silencing these pathways during different phases of task acquisition (during the task vs after the task), the authors report valuable findings of the involvement of these cerebellar pathways in learning and consolidation.Strengths:The experiments are well-executed. The authors perform multiple controls and careful analysis to solidly rule out any gross motor deficits caused by their cerebellar nuclei manipulation. The finding that cerebellar projections to the thalamus are required for learning and execution of the accelerating rotarod task adds to a growing body of literature on the interactions between the cerebellum, motor cortex, and basal ganglia during motor learning. The finding that silencing the cerebellar nuclei after a task impairs the consolidation of the learned skill is interesting.

We thank the reviewer for the positive comments and insightful critics below.

Weaknesses:While the controls for a lack of gross motor deficit are solid, the data seem to show some motor execution deficit when cerebellar nuclei are silenced during task performance. This deficit could potentially impact learning when cerebellar nuclei are silenced during task acquisition.

One of our key controls are the tests of the treatment on fixed speed rotarod, which provides the closest conditions to the ones found in the accelerating rotarod (the main difference between the protocols being the slow steady acceleration of rod rotation in the accelerating version). Indeed, small but measurable deficits are found at the highest speed in the fixed speed rotarod in the CN-VAL group, while there was no measurable effect on the CN-CL group, which actually shows lower performances from the second day of learning; we believe this supports our claim that the CN-CL inhibition impacted more the learning process than the motor coordination. In contrast, the CN-VAL group only showed significantly lower performance on day 4 consistent with intact learning abilities. Yet, under CNO, CN-VAL mice could stay for more than a minute and half at 20rpm, while in average they fell from the accelerating rotarod as soon as the rotarod reached the speed of ~19rpm (130s). Overall, we focused our argument on the first days of learning where the differences between the groups are more pronounced. We clarified the discussion (section “A specific impact on learning of CL-projecting CN neurons.”)

Separately, I find the support for two separate cerebello-thalamic pathways incomplete. The data presented do not clearly show the two pathways are anatomically parallel. The difference in behavioral deficits caused by manipulating these pathways also appears subtle.

There is indeed a paragraph devoted to the discussion of this point (last part of the section “A specific impact on learning of CL-projecting CN neurons.”). Briefly, we actually know from the literature that there is a degree of collateralization (CN neurons projecting to both VAL and CL, see refs cited above), but it does not seem logically possible that the exact same population would have different effects, which are very distinct during the first learning days. The only possible explanation is the CN-CL and CN-VAL infections recruit somewhat different populations of neurons. We have now added more experiments to support our finding using retrograde infections using two rAAV viruses expressing red and green fluorescent reporter. These experiments confirm the limited overlap of the two populations of interest obtained by retrograde infection. We feel thus confident that while some CN neurons may project to both structures, retrograde infection strategies thus appear to differentially infect CN populations.

While we agree that after 3-4 days of learning the difference between the groups becomes elusive, we respectfully disagree with the reviewer that in the early stages these differences are negligible.

**Reviewer #3 (Public review):**
Summary:Varani et al present important findings regarding the role of distinct cerebellothalamic connections in motor learning and performance. Their key findings are that:(1) Cerebellothalamic connections are important for learning motor skills(2) Cerebellar efferents specifically to the central lateral (CL) thalamus are important for shortterm learning(3) Cerebellar efferents specifically to the ventral anterior lateral (VAL) complex are important for offline consolidation of learned skills, and(4) That once a skill is acquired, cerebellothalamic connections become important for online task performance.The authors went to great lengths to separate effects on motor performance from learning, for the most part successfully. While one could argue about some of the specifics, there is little doubt that the CN-CL and CN-VAL pathways play distinct roles in motor learning and performance. An important next step will be to dissect the downstream mechanisms by which these cerebellothalamic pathways mediate motor learning and adaptation.Strengths:(1) The dissociation between online learning through CN-CL and offline consolidation through CN-VAL is convincing.(2) The ability to tease learning apart from performance using their titrated chemogenetic approach is impressive. In particular, their use of multiple motor assays to demonstrate preserved motor function and balance is an important control.(3) The evidence supporting the main claims is convincing, with multiple replications of the findings and appropriate controls.

We thank the reviewer for the positive comments and insightful critics below.

Weaknesses:(1) Despite the care the authors took to demonstrate that their chemogenetic approach does not impair online performance, there is a trend towards impaired rotarod performance at higher speeds in Supplementary Figure 4f, suggesting that there could be subtle changes in motor performance below the level of detection of their assays.

This is now better acknowledged in the discussion in the section “A specific impact on learning of CL-projecting CN neurons.” However, we want to underline that the strongest deficit in learning is found in animals with CN->CL inhibition which latency to fall saturates at about 100s on the rotarod; this indicates that mice fall as soon as the accelerating rotarod speed reaches about 16rpm. In fixed speed rotarod, the inhibition of CN->CL neurons shows not even a trend of difference at 15rpm with control mice, and the animals run 2 minutes without falling at this speed. This makes us confident that the CN->CL pathway interfers more with the learning than with the actual locomotor function on the rotarod.

(2) There is likely some overlap between CN neurons projecting to VAL and CL, somewhat limiting the specificity of their conclusions.

This issue is treated in the discussion. (see also replies to reviewers 1 and 2 above). We added experiments with simultaneous retro-AAV infections in CL and VAL and the data are presented in Supplementary Figure 5. We found that retrograde infection targeted different populations of CN neurons; although collaterals in both CL and VAL may be present for (some of) these two populations of neurons, they are likely strongly biased toward one or the other thalamic regions, explaining the differential retrograde labelling in the CN. We hope these experiments will answer the reviewer’ s concern.

**Recommendations for the authors:**

**Reviewer #2 (Recommendations for the authors):**
(1) Multiple studies have reported on the effect of cerebellar nuclei (CN) manipulation on locomotion. Here the authors perform several controls and careful analysis to rule out gross motor deficits caused by DREADD-mediated CN silencing. As the authors point out in the discussion, part of the difference from prior studies could be the mild degree of inhibition here. However, it is possible that the CN inhibition here induces a subtle motor deficit and the accelerating rotarod task is challenging and more readily reveals this motor deficit, rather than a deficit in motor learning per se. Two pieces of data seem to suggest this:(a) under CN inhibition during the task (Figure 1i), mice could never achieve the level of performance as mice under CN inhibition after the task, even after several days of training, which suggests the CN inhibition is interfering with task performance;(b) in highly trained mice (after learning), applying the CN inhibition impaired performance to a similar extend as mice in Figure 1i (Figure 4).Can the authors rule out the possibility that CN inhibition during the task is impairing motor execution rather than motor learning?

We do not rule out a contribution of impaired motor coordination at the highest speed (last paragraph of the section “A specific impact on learning of CL-projecting CN neurons.”). Indeed, most of our argument in favor of deficit in learning is primarily in the first days (Early phase), particularly for the CN->CL CNO group (Fig 3h). A crucial control in our work is the use of fixed speed rotarod, where no deficit is observed. The difference between the fixed and accelerating rotarod is rather minimal since the acceleration of the rotarod is rather small (0.12rpm/s for speed up to >20 rpm).

Interpreting the effect of treatment reversal is challenging. If the only effect of CNO was a motor deficit, the animals who learned under CNO should rapidly regain higher performance under saline, which is not observed. When switching from CNO to Saline after 7 days of training, it is difficult to disentangle which part is due to a crude motor deficit (which would not show in fixed speed rotarod), and which part is due to an unability to resume motor learning after the task has been (mis-)consolidated.

(2) The separation of the cerebellar pathways to the intralaminar thalamus (IL) and ventral thalamus (VAL) is not clear to me. It is not clear the CN neurons projecting to these nuclei are distinct. In addition, although IL projects to the striatum and VAL does not, both IL and VAL project to motor cortex. It is unclear to what extent these pathways can be separated. The argument for distinct pathways (as laid out in the discussion) is the distinct behavior deficits when manipulating these two pathways, but this difference seems subtle (point 3).

We now clarify that CN populations are different help to retrograde labelling experiments (new Suppl Fig 5). A discussion on the differences in IL and VAL projections is now discussed in the last paragraph of the section “A specific impact on learning of CL-projecting CN neurons.” Briefly, we argue that the despite some overlap of their targets, the profiles of the CL and VAL differ substantially.

(3) The pattern of behavioral deficits induced by CN->CL and CN->VAL neurons appear similar in Figure 3b-c and e-f. I have difficulty seeing how these data lead to the differences in the regression fits in panels 3g-k, which seem to show distinct patterns of performance change within and across sessions. One notable difference in Figure 3b-c and e-f seems to be that CN->VAL CNO treated mice exhibit lower performance on the very first trial for most days. Somehow, this pattern is present even after the CNO treatment is switched to saline (Figure 3f). I wonder if this data point is driving the difference. One control analysis the authors could do is to exclude the 1st trial and test if the effects are preserved.

Since the learning is cumulative and involves varying degree of consolidation it is indeed difficult to substantiate the difference from the average performance: a performance on day 3 may be limited by slow learning and perfect consolidation or good learning and imperfect consolidation. That is why we designed an analysis which takes into account the observed relationships between initial performance, within session gain of performance and acrosssession carry-over of this gain of performance (Fig 2). This analysis focuses on the first days of learning, before the performance plateau is reached in the CNO groups. While a clear deficit in consolidation is observed with full CN inhibition, this is not the case for the CN→CL CNO groups, despite their weaker performance after 3 days, similar to that seen with full CN inhibition. In contrast, normal learning is observed in the CN→VAL CNO group during these three days. The consolidation deficit in the CN→VAL CNO group is more subtle than in the CN CNO group and is indeed largely driven by the first data point. This is consistent with the idea that CN→VAL inhibition only partially impairs consolidation (compared to full CN inhibition), leaving some “savings” that allow rapid reacquisition.

(4) The quantification of locomotion in Figure S2 needs more information. What is linear movement? What is sigma? What is the alternation coefficient? These are not defined in the legends or the Methods as far as I can tell. Related to point 1 above, the authors should provide some analysis of the stride length and hindlimb to forelimb distance as measures of locomotion execution.

These measures were taken from Simon J Neurosci 2004 24(8):1987-1995 which is now cited and their description is now provided in the Methods.

Minor:(5) To help readers follow the logic of experimental design, please explain why CNO was switched to saline after day 4 in Figures 1j, 3c, and f. Specifically, is the saline manipulation meant to test something as opposed to applying CNO throughout the entire course of the behavioral test?

Since we had no difference between the groups at the end of the Early phase, we decided to test whether the skill consolidated under CNO remained available when the CNO was removed (and it indeed was). This is now more clearly stated in the Results.

(6) I have difficulty understanding what is plotted in Figure 4b and d. The legend says the change in performance is calculated the same way as in Figure 2a, so the changes are presumably the regression slopes. But how are the regression slopes calculated for daily start (1st trial) and daily end (last trial)?

Skill level at the beginning and end of each trial correspond to the values of the regression line for abscissae values of trial 1 and trial 7 (green points). This has been added to the figure legend.

(7) Do CN-CL and CN-VAL neurons also project to other brain regions besides the thalamus? Might these pathways also contribute to learning and consolidation of the accelerating rotarod task? Please discuss.

This is now discussed in more detail in the last paragraph of the section “A specific impact on learning of CL-projecting CN neurons.”

**Reviewer #3 (Recommendations for the authors):**
(1) Please check the anatomic evidence for the strict dichotomy between intralaminar (specifically central lateral nucleus) nuclei projecting to the striatum and the ventral-anteriorlateral (VAL) complex projecting to the cortex. For example, while the Chen et al paper shows that there are cerebellar-intralaminar-striatal projections, it does not exclude intralaminar cortex projections, which have at least been demonstrated in rats. Similarly, VAL has projections to striatum (see, e.g., Smith et al, "The thalamostriatal system in normal and diseased states", Frontiers in Systems Neuroscience, 2014). It may be that some of these projections are stronger, but I don't think it's true that these pathways are as well-separated as the authors suggest. I also don't think this changes the fundamental conclusions but is important for potential mechanisms by which differential learning could occur and necessitate modification of Figure 5.

We have toned down the interpretation of CL and VAL relaying specifically to different brain structures and mostly put forward the duality of the pathways. The connections with the cortex are now discussed at the end of the section “A specific impact on learning of CL-projecting CN neurons.”

(2) Please provide more details on the spike sorting. By what metrics were single units declared to be well-separated? How many units were identified under each condition? What was the distribution of firing rates with and without CNO treatment? Are the units shown in panel 1f from before and after CNO as in panel E or are just 2 examples of isolated units? The units by themselves are not very helpful to the reader. Showing sample auto and/or crosscorrelograms for units recorded on the same electrode would be more helpful to show how well-isolated the units are.

Single units were considered well-isolated based on quantitative quality metrics computed after MountainSort 4 spike sorting (Phyton 3.8). Units were required to have a signal-to-noise ratio (SNR) greater than 5, inter-spike interval (ISI) violations less than 1%, an amplitude cutoff below 0.1, a presence ratio above 0.9, a firing rate greater than 0.1 Hz, and at least 50 detected spikes. In addition, units were assessed for temporal stability across the recording using autocorrelograms and presence over the recording, ensuring there were no prolonged periods of total inactivity. Units meeting these criteria were deemed well-separated and reliable for further analysis. This has been added to the Methods.

Cell numbers are provided with the statistics in the supplementary table for fig panel 1g. Panels are from the same unit before and after CNO. Example of auto- crosscorr- are provided in the new Supplementary Figure 6.

(3) Panel 2g - "firing rate modulation" is unclear. I think the authors are showing the mean firing rate with DREADD+CNO treatment divided by the mean firing rate in the pre-CNO condition for the same group (I couldn't find that in the Methods, my apologies if I missed it)? However, firing rate modulation to me means variability in firing rate within a recording. Perhaps "relative firing rate" or "% pre-CNO firing rate" would be clearer?

The definition has been added to the Method and the axis has been changed to ‘Change in FR induced by SAL/CNO’

(4) Figure 3f - why does consolidation appear to be impaired after the transition from CNO to saline between sessions, when in panel 1j suppressing the CN does not have a similar effect once CNO is switched to saline? Could this be driven by a small number of mice? Since a central conclusion of the paper is that CN-VAL connections are uniquely important for posttraining consolidation, this discrepancy is important to explain - if the results post-saline are spurious, how do we know that the results post-CNO aren't also spurious? Panels similar to Figure 4b and d showing all the data from the last/first trial of each session I think would be convincing.

Our results overall indicate that the overnight consolidation of the improvement in performance seem only effective in the early phase (as pointed out on the summary figure 5). We do not believe then that the saline results are spurious.

It can be seen indeed in the control groups of the figure 1; to make this more visible, we plot in Author response image 1 the difference between trial 7 and trial 1 the next day. An overnight drop in performance becomes visible in the late phase.

The decrement on the first trial in the first 3 days is visible for the majority of the mice. The plot asked by the reviewer is represented in the Author response image 2.

**Author response image 2. sa3fig2:** 

Minor points:(5) In panel 1a, the solid yellow line obscures a lot of the image and I don't think adds anything.

We assume this was referring to a line on fig1d, which has been removed.

(6) Panel 2a - color selection could present problems for those with red-green color blindness.

This has been fixed.

(7) Supplementary Figure 3 - what are the arrows and arrowheads indicating?

These have been removed.

(8) In the Discussion: "Studies of cerebellar synaptic plasticity provide clearly support the involvement of cerebellum in rotarod learning..." Delete the word "provide"

This has been fixed

(9) "This indicates that either the distinct functional roles of VAL-projecting or CLprojecting." The second "of" should be "or", I think.

This has been fixed.